# CROSS-TRAJECTORY REPRESENTATION LEARNING FOR ZERO-SHOT GENERALIZATION IN RL

**Bogdan Mazoure**[*†]
bogdan.mazoure@mail.mcgill.ca
McGill University, Quebec AI Institute

**Ahmed M. Ahmed**[*†]
ahmedah@stanford.edu
Stanford University

**Patrick MacAlpine**[†]
patrick.macalpine@sony.com
Sony AI

**R Devon Hjelm**
devon.hjelm@microsoft.com
Université de Montréal, Quebec AI Institute,
Microsoft Research

**Andrey Kolobov**
akolobov@microsoft.com
Microsoft Research

## ABSTRACT

A highly desirable property of a reinforcement learning (RL) agent – and a major difficulty for deep RL approaches – is the ability to generalize policies learned on a few tasks over a high-dimensional observation space to similar tasks not seen during training. Many promising approaches to this challenge consider RL as a process of training two functions simultaneously: a complex nonlinear *encoder* that maps high-dimensional observations to a latent *representation* space, and a simple linear *policy* over this space. We posit that a superior encoder for zero-shot generalization in RL can be trained by using solely an auxiliary SSL objective if the training process encourages the encoder to map *behaviorally similar* observations to similar representations, as reward-based signal can cause overfitting in the encoder (Raileanu and Fergus, 2021). We propose Cross Trajectory Representation Learning (CTRL), a method that runs within an RL agent and conditions its encoder to recognize behavioral similarity in observations by applying a novel SSL objective to pairs of trajectories from the agent's policies. CTRL can be viewed as having the same effect as inducing a pseudo-bisimulation metric but, crucially, avoids the use of rewards and associated overfitting risks. Our experiments[1] ablate various components of CTRL and demonstrate that in combination with PPO it achieves better generalization performance on the challenging Procgen benchmark suite (Cobbe et al., 2020).

## 1 INTRODUCTION

Deep reinforcement learning (RL) has emerged as a powerful tool for building decision-making agents for domains with high-dimensional observation spaces, such as video games (Mnih et al., 2015), robotic manipulation (Levine et al., 2016), and autonomous driving (Kendall et al., 2019). However, while deep RL agents may excel at the specific task variations they are trained on, learning behaviors that generalize across a large family of similar tasks, such as handling a variety of objects with a robotic manipulator, driving under a variety of conditions, or coping with different levels in a game, remains a challenge. This problem is especially acute in zero-shot generalization (ZSG) settings, where only a few sequential tasks are available to learn policies that are meant to perform well on different yet related tasks without further parameter adaptation. ZSG settings highlight the fact that generalization often cannot be solved by more training, as it can be too expensive or impossible to instantiate all possible real-world deployment scenarios a-priori.

---

[*]Equal contribution. [†]The author did part of the work for this paper while at Microsoft.
[1]Code link: https://github.com/bmazoure/ctrl_public

In this work, we aim to improve ZSG in RL by proposing a new way of training the agent's *representation*, a low-dimensional summary of information relevant to decision-making extracted from the agent's high-dimensional observations. Outside of RL, representation learning can help with ZSG, e.g. using unsupervised learning to obtain a representation that readily transfers to unseen classes in vision tasks (Bucher et al., 2017; Sylvain et al., 2020; Wu et al., 2020). In RL, unsupervised representation learning in the form of auxiliary objectives can be used to provide a richer learning signal over learning from reward alone, which helps the agent avoid overfitting on task-specific information (Raileanu and Fergus, 2021). However, to our knowledge, no unsupervised learning method used in this way in RL has thus far been shown to substantially improve performance in ZSG over end-to-end reward-based methods (e.g., Cobbe et al., 2020).

We posit that using unsupervised (reward-free) learning to find representations that capture behavioral similarity across different trajectories will improve ZSG in RL. We note that the bisimulation framework (Ferns et al., 2004) does this directly with rewards, optimizing an agent to treat states as behaviorally similar based on the expected reward, and this has been shown to help in visual generalization settings (Zhang et al., 2021). We expand on this framework to improve ZSG performance, using unsupervised learning to train an agent that recognizes behavior similarity in a reward-free fashion. To do so, we propose *Cross Trajectory Representation Learning (CTRL)*, which applies a novel self-supervised learning (SSL) objective to pairs of trajectories drawn from the agent's policies. For optimization, CTRL defines a prediction objective across trajectory representations from nearby partitions defined by an online clustering algorithm. The end result is an agent whose encoder maps behaviorally similar trajectories to similar representations without directly referencing reward, which we show improves ZSG performance over using pure RL or RL in conjunction with other unsupervised or SSL methods.

Our main contributions are as follows:

- We introduce Cross Trajectory Representation Learning (CTRL), a novel SSL algorithm for RL that defines an auxiliary objective across trajectories in order to capture the notion of *behavioral similarity* in the representations of the agent's belief states. CTRL's approach is two-fold: (i) it uses a clustering loss to group representations of behaviorally similar trajectories and (ii) boosts cross-predictivity between trajectory representations from nearby clusters.

- We empirically show that CTRL improves zero-shot generalization in the challenging Procgen benchmark suite (Cobbe et al., 2020). Through a series of ablations, we highlight the importance of cross-trajectory views in boosting behavioral similarity.

- We connect CTRL to the class of bisimulation methods, and provide sufficient conditions under which both formalisms can be equivalent.

## 2 BACKGROUND, MOTIVATION, AND RELATED WORKS

There are a broad class of ZSG settings in RL, such as generalization across reward functions (Barreto et al., 2016; Touati and Ollivier, 2021; Misra et al., 2020), observation spaces (Zhang et al., 2021; Li et al., 2021; Raileanu and Fergus, 2021), or task dynamics (Rakelly et al., 2019). For each of these settings, there are a number of promising directions for improving ZSG performance: giving the agent better exploration policies (Van Roy and Wen, 2016; Misra et al., 2020; Agarwal et al., 2020), meta learning (Oh et al., 2017; Gupta et al., 2018; Rakelly et al., 2019), or planning (Sohn et al., 2018). In this work, we focus on directly improving the agent's *representations*. The agent's representations are high-level abstractions of observations or trajectories from the environment (e.g., the output of an encoder), and the desired property here is that one can easily learn a policy on top of that representation such that the combined model (i.e., the agent) generalizes to novel situations. The tasks are assumed to share a common high-level goal and are set in environments that have the same dynamics, but each task may need to be accomplished under different initial conditions and may differ visually. As the policy is built upon the agent's representations, this motivates the focus of this work for improving generalization: unless the agent's representations generalize well, one cannot expect its policy to readily do so.

Unsupervised representation learning has been shown to improve generalization across domains, including zero-shot in vision (Sylvain et al., 2020; Wu et al., 2020) and sample-efficiency in RL (Ey-

senbach et al., 2019; Schwarzer et al., 2021; Stooke et al., 2021). In RL, unsupervised objectives can be used as an *auxiliary objective* (or auxiliary task, Jaderberg et al., 2017), which provide an alternative signal to reward-based learning signal. Due to the potential role the RL loss may play in overfitting (Raileanu and Fergus, 2021), we believe that having a learning objective for agent's representation that is separate from that of its policy is crucial for good ZSG performance.

**Self-supervised learning and reinforcement learning.** A successful class of models that incorporate unsupervised objectives to improve RL use self-supervised learning (SSL) (Anand et al., 2019; Srinivas et al., 2020; Mazoure et al., 2020; Schwarzer et al., 2021; Stooke et al., 2021; Higgins et al., 2017). SSL formulates objectives by generating different *views* of the data, which are essentially transformed versions of the data, e.g., generated by using data augmentation or by sampling patches. While successful in their own way, prior works that combine SSL with RL do so by applying known SSL algorithms (e.g., from vision, Hjelm et al., 2018; van den Oord et al., 2019; Bachman et al., 2019; Chen et al., 2020; He et al., 2020; Grill et al., 2020) to RL in a nearly off-the-shelf manner, predicting state representations *within* a given trajectory, only potentially using other trajectories as counterexamples in a contrastive loss. As such, these methods' representations can have trouble generalizing latent behavioral patterns present in ostensibly different trajectories.

**Bisimulation metrics in reinforcement learning.** Our hypothesis is that ZSG is achievable if the agent recognizes behavioral similarity between trajectories based on their long-term evolution. Learning this sort of behavior similarity is a central characteristic of bisimulation metrics (Ferns et al., 2004), which assign a value of 0 to states which are behaviorally indistinguishable and have the same reward. Reward-based bisimulation metrics have been shown to learn representations that have a number of useful properties, e.g.: smoothness (Gelada et al., 2019), visual invariance (Zhang et al., 2021), action equivariance (van der Pol et al., 2020) and multi-task adaptation (Zhang et al., 2020). For ZSG however, encoding relational information based on reward may not actually help (Misra et al., 2020; Touati and Ollivier, 2021; Yang and Nachum, 2021; Agarwal et al., 2021), as the agent may overfit to spurious correlations between high-dimensional observations and the reward signal seen during training. HOMER (Misra et al., 2020) expands on the concept of bisimulation to learn behavioral similarity between states using unsupervised exploration at deployment. Among the existing methods, PSEs (Agarwal et al., 2021) reward-free notion of behavioral similarity is conceptually the closest to CTRL's, and we compare these algorithms empirically in Section 6. However, algorithmically and in terms of their modes of operation, CTRL and PSE are very different. PSE assumes the availability of expert policies for training tasks and learns a representation using trajectories from these experts and an action distance measure, which it also assumes to be provided. CTRL doesn't make these assumptions and learns a representation online from trajectories simultaneously generated by its substrate RL algorithm.

**Mining views across unsupervised clusters.** Given our hypothesis that a model that learns behavioral similarities using signal other than reward will perform well on ZSG, there are still many potential models available to learn said similarities in an unsupervised way. A simple and natural choice is to collect agent trajectories as examples of behaviors, then do clustering (online, similar to Asano et al., 2020; Caron et al., 2020) over trajectories. In RL, Proto-RL (Yarats et al., 2021) also uses clustering to obtain a pre-trained set of prototypical states, but for a different purpose – to estimate state visitation entropy in hard exploration problems. However, clustering alone may not be sufficient to recognize behaviors necessary for ZSG, as representations built on clustering only need to partition behaviors, which may bias the model towards similarities evident training experience. This would be counter-productive for our generalization goal. We therefore use a second objective built on top of the structure provided by clustering to learn a more diverse set of similarities. Drawing inspiration from Mine Your Own View (MYOW, Azabou et al., 2021), CTRL selects (*mines*) representational nearest neighbors from different, nearby clusters and applies a predictive SSL objective to them. This cross-cluster objective encourages CTRL to recognize a larger set of similarities than would be necessary to cluster on the training set, which we show improves ZSG performance.

## 3 PROBLEM STATEMENT AND PRELIMINARIES

Formally, we define our problem setting w.r.t. a discrete-time Markov decision process (MDP) $M \triangleq \langle \mathcal{S}, \mathcal{A}, \mathcal{P}, \mathcal{R} \rangle$, where $\mathcal{S}$ is a state space, $\mathcal{A}$ is an action space, $\mathcal{P} : \mathcal{S} \times \mathcal{A} \times \mathcal{S} \rightarrow [0, 1]$ is a

transition function characterizing environment dynamics, and $\mathcal{R} : \mathcal{S} \times \mathcal{A} \to \mathbb{R}$ is a reward function. $M$'s state and action spaces may be discrete or continuous, but in the rest of the paper we assume them to be discrete to simplify exposition. In practice, an agent usually receives observations but not the full information about the environment's current state. Consider an observation space $\mathcal{O}$ and an observation function $\mathcal{Z} : \mathcal{S} \times \mathcal{O} \to [0, 1]$ that define what observations an agent may receive and how these observations are generated (possibly stochastically) from $M$'s states. We define a *task $T$* as a partially observable MDP (POMDP) $T = \langle \mathcal{S}, \mathcal{A}, \mathcal{P}, \mathcal{R}, \mathcal{O}, \mathcal{Z}, s_0 \rangle$, where $s_0 \in \mathcal{S}$ is an initial state. Although many RL agents make decisions in a POMDP based only on the current observation $o_t$ or at most a few recent ones, in general this may require using information from the entire observation history $o_1, \ldots, o_t$ so far. Denoting the space of such histories as $\mathcal{H}$, computing an agent's behavior for task $T$ amounts to finding a policy $\pi : \mathcal{H} \times \mathcal{A} \to [0, 1]$ with the optimal or near-optimal expected return from the initial state $V_T^\pi \triangleq \mathbb{E} \left[ \sum_{t=0}^\infty \gamma^t \mathcal{R}(S_t, \pi(H_t) \mid S_0 = s_0 \right]$, where $S_t$ and $H_t$ are random variables for the POMDP's underlying state and agent's observation history at time step $t$, respectively, and $\gamma$ is a discount factor. For an MDP $M = \langle \mathcal{S}, \mathcal{A}, \mathcal{P}, \mathcal{R} \rangle$, a set $\mathscr{O}$ of observation spaces, and a set $\mathscr{Z}$ of observation functions w.r.t. $\mathcal{S}$, let a *task family* be the POMDPs set $\mathscr{T}_{M,\mathscr{O},\mathscr{Z}} \triangleq \{ \langle \mathcal{S}, \mathcal{A}, \mathcal{P}, \mathcal{R}, \mathcal{O}, \mathcal{Z}, s_0 \rangle \}_{\mathcal{O} \in \mathscr{O}, \mathcal{Z} \in \mathscr{Z}, s_0 \in \mathcal{S}}$. We assume that different observation spaces in $\mathscr{O}$ have the same mathematical form, e.g., pixel tensors representing possible camera images, but correspond to qualitatively distinct subspaces of this larger space, such as subspaces of images depicting brightly and dimly lit scenes.

Our training and evaluation protocol is formalized w.r.t. a *task distribution* $d(\mathscr{T}_{M,\mathscr{O},\mathscr{Z}}) \triangleq P(\mathscr{O}, \mathcal{Z}, \mathcal{S})$ *over task family* $\mathscr{T}_{M,\mathscr{O},\mathscr{Z}}$, where $P(\mathscr{O}, \mathcal{Z}, \mathcal{S})$ is a joint probability mass over observation spaces, observation functions, and initial states. In the rest of the paper, $M$, $\mathscr{O}$, and $\mathcal{Z}$ will be clear from context, and we will denote the task family as $\mathscr{T}$ and the task distribution as $d(\mathscr{T})$. For agent training, we choose $N$ tasks from $\mathscr{T}$, and denote the distribution $d(\mathscr{T})$ restricted to these $N$ tasks as $d(\mathscr{T}_N)$. During the training phase, the RL agent learns via a series of epochs (themselves composed of episodes), by sampling a task from $d(\mathscr{T})$ independently at the start of each episode, until the total number of time steps exceeds its training budget. In each epoch, an RL algorithm uses a batch of trajectories gathered from episodes in order to compute gradients of an RL objective and update the parameters of the agent's policy $\pi$. In representation learning-aided RL, a policy is viewed as a composition $\pi = \theta \circ \phi$ of an *encoder* $\phi : \mathcal{H} \to \mathcal{E}$ and a *policy head* $\theta : \mathcal{E} \times \mathcal{A} \to [0, 1]$, both of which are the outputs of the training phase. The training phase is followed by an evaluation phase, during which the policy is applied to tasks sampled from $d(\mathscr{T} \setminus \mathscr{T}_N)$, distribution $d$ restricted to the set of tasks $\mathscr{T} \setminus \mathscr{T}_N$ not seen during training. *Our focus on generalization means that we seek a policy $\pi$ whose encoder $\phi$ allows it to maximize* $\mathbb{E}_{T \sim d(\mathscr{T} \setminus \mathscr{T}_N)}[V_T^\pi]$ *despite being trained only on distribution $d(\mathscr{T}_N)$.*

## 4 ALGORITHM

CTRL's key conceptual insight is that capturing reward-agnostic behavioral similarity improves ZSG, because it enables $\phi$ to correctly associate previously unseen observation histories with those for which the agent's RL-trained behavior prescribes a good action. CTRL runs synchronously with an online RL algorithm, which is crucial to ensure that as the agent's policies improve, so does the notion of behavioral similarity induced by CTRL.

Like most online RL methods themselves, our algorithm operates in epochs, learning from a batch of trajectories in each epoch. CTRL assumes that all trajectories within each of its training batches come from the same policy. Before the RL algorithm updates the policy head in a given epoch, CTRL uses a trajectory batch from the current policy to update the encoder with gradients of a novel auxiliary loss $\mathcal{L}_{\text{CTRL}}$ that we describe in this section.

### 4.1 INTUITION AND HIGH-LEVEL DESCRIPTION

**Algorithm overview.** For each trajectory batch, CTRL performs 4 operations:

1. Apply the observation history (belief state) encoder $\phi$ to generate a low-dimensional reward-agnostic representation ("view") of each trajectory.
2. Group trajectories' views into $C$ sets ($C$ is a tunable hyperparameter) using an online clustering algorithm with loss $\mathcal{L}_{\text{clust}}$ (Equation (4), Section 4.2).

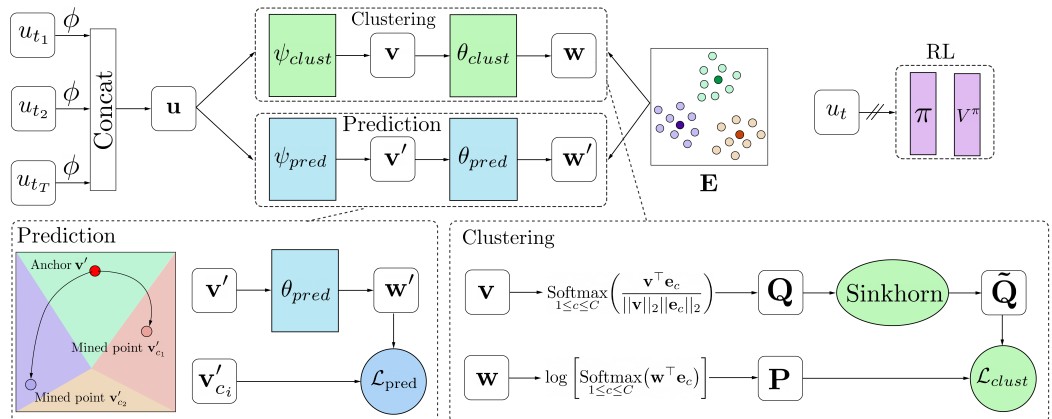

Figure 1: Schematic view of CTRL's key steps for every trajectory batch. **(i) Generating trajectory views** *(top left)*. For each trajectory in a batch, CTRL samples a subsequence of its time steps, computes belief-state/action embeddings $u_{t_i}$ with encoder $\phi$, and concatenates them into a trajectory representation (view) $\mathbf{u}$. **(ii) Clustering trajectory views** *(bottom right)*. CTRL uses the online Sinkhorn-Knopp clustering procedure (Caron et al., 2020): for each trajectory view $u$, it produces two new views $\mathbf{v}$ and $\mathbf{w}$, soft-clusters all trajectories' $\mathbf{v}$s and $\mathbf{w}$s into $C$ clusters, and uses a measure of consistency between these two clusterings as a loss $\mathcal{L}_{\text{clust}}$. In the diagram, variables $\mathbf{e}_c$ denote cluster centroids. **(iii) Encouraging cross-cluster behavioral similarity** *(bottom left)*. After computing trajectory view clusters, CTRL applies a variant of MYOW (Azabou et al., 2021) to them. Namely, it repeatedly samples a trajectory view $\mathbf{v}'$, computes a new view $\mathbf{w}'$ for it, and computes a loss $\mathcal{L}_{\text{pred}}$ that penalizes differences between $\mathbf{w}'$ and views $\mathbf{v}'_{\mathbf{c_i}}$ of randomly chosen trajectories from $\mathbf{v}'$'s neighboring clusters. Encoder $\phi$ and auxiliary predictors used by CTRL are then updated using $\mathcal{L}_{\text{CTRL}} = \mathcal{L}_{\text{clust}} + \mathcal{L}_{\text{pred}}$'s gradients *(top right)*.

3. Using trajectory pairs selected from neighboring clusters, apply a predictive loss $\mathcal{L}_{\text{pred}}$ (Equation (6), Section 4.2) to encourage $\phi$ to capture cross-cluster behavioral similarities.

4. Update $\phi$ with gradients of the total loss: $\mathcal{L}_{\text{CTRL}} = \mathcal{L}_{\text{clust}} + \mathcal{L}_{\text{pred}}$.

The schema in Figure 1 provides a high-level outline of these steps' implementation and explains their interplay within CTRL, accompanied by an intuition for each step (below) a more detailed description in Section 4.2. We conduct ablations to show the effect of removing the clustering and predictive objectives of CTRL, with details in Appendix 8.3.

**Clustering.** $C$ clusters can be viewed as corresponding to $C$ latent "situations" in which an RL agent may find itself. Each situation is essentially a group of belief states. CTRL's implicit hypothesis is that a given policy should behave roughly similarly across all belief states corresponding to the same "situation", i.e., generalize across similar belief states. Under this hypothesis, an agent's policy can be expected to produce $C$ sets of roughly similar trajectories. CTRL's clustering step (#2 above) is an attempt to recover these trajectory sets. Since each trajectory consists of belief states, the purpose of $\mathcal{L}_{\text{clust}}$ is to force the encoder $\phi$ to compute belief state representations that make trajectories within each cluster look similar in the latent space.

Since we would like to evolve clustering online as new trajectory batches arrive, we employ a common online clustering algorithm, the Sinkhorn-Knopp procedure (Caron et al., 2020), which has been used as an auxiliary RL loss (e.g., Proto-RL, Yarats et al., 2021).

**Cross-cluster prediction.** Note, however, that the clustering loss emphasizes the recognition of behavioral similarities *within* clusters. This may hurt generalization, as the resulting centroids may not faithfully represent behaviors encountered at test time. Our hypothesis is that encouraging encoder $\phi$ to induce latent-space similarities between trajectories from *different but adjacent* clusters will increase its ability to recognize behaviors in unseen test trajectories.

While there are several ways to encourage cross-cluster representational similarity, using a mechanism similar to MYOW (Azabou et al., 2021) on trajectories drawn from neighboring clusters captures this idea particularly well. Namely, to get the cross-predictive loss $\mathcal{L}_{\text{pred}}$, we sample trajectory view pairs from neighboring clusters and apply the cosine-similarity loss to those pairs.

**Using reward guidance without reward signal for representation learning.** CTRL trains encoder $\phi$ only using the gradients of $\mathcal{L}_{\text{CTRL}}$; the RL algorithm's loss $\mathcal{L}_{\text{RL}}$ trains only the policy head. Thus, encoder $\phi$ is isolated from the previously observed dangers of overfitting to the reward function (Raileanu and Fergus, 2021) that shapes $\mathcal{L}_{\text{RL}}$. However, we emphasize that CTRL's representation learning is nonetheless very much guided by the reward function, although indirectly: the training batches of belief state and action trajectories are still collected from policies learned by the policy head via $\mathcal{L}_{\text{RL}}$'s gradients, which are reward-dependent.

## 4.2 DETAILS

Below we describe the details of each of CTRL's steps, with CTRL's pseudocode presented in **Algorithm 1 in Appendix 8.1**. While in general the agent's belief state at step $t$ of a trajectory is the entire observation history $h_t = (o_1, \ldots, o_t)$, in the rest of the section we will assume $h_t = (o_t)$ and, in a slight abuse of notation, use $\phi(o_t)$ instead of $\phi(h_t)$ to simplify explanations[2]. We emphasize, however, that CTRL equally applies in settings where the agent uses a much longer history of observations as its state. In this case, $\phi$ would be recurrent or process stacks of frames.

We also note that our CTRL implementation's high-level algorithmic choices for clustering and cross-cluster prediction – Sinkhorn-Knopp and MYOW, respectively – come from prior works (Caron et al., 2020; Azabou et al., 2021).

**Generating low-dimensional trajectory views with encoder $\phi$.** CTRL's input in each epoch is a trajectory batch $\{traj_i\}_{i=1}^{B}$ of size $B$. Assume all trajectories in the batch have the same length $L$. For an integer hyperparameter $T \leq L$, for each trajectory $traj_i = (o_0, a_0, r_0, \ldots o_{T_b}, a_L, r_L)$ we independently and uniformly sample a subset of its steps $\tau_i = t_1, \ldots, t_T$ to form a subtrajectory $(o_{t_1}, a_{t_1}, r_{t_1}, \ldots o_{t_T}, a_{t_T}, r_{t_T})$. We then encode this subtrajectory as

$$\boldsymbol{u}_i^{(\tau_i)} = (FiLM(\phi(o_{t_1}), a_{t_1}), .., FiLM(\phi(o_{t_T}), a_{t_T})), \tag{1}$$

where $FiLM(\phi(o_{t_j}), a_{t_j})$ is a common way of combining representations of different objects, akin to conditioning Perez et al. (2018), and the resulting vector $\boldsymbol{u}_i^{(\tau_i)}$ is in a low-dimensional space $\mathcal{U}$. Note two aspects of the process of generating these vectors: (1) it drops rewards from the original trajectory and (2) it critically relies on $\phi$ whose parameter values are learned from previous epochs. Vectors $\boldsymbol{u}_i^{(\tau_i)}$ produced in this way are the trajectory views that the next steps of CTRL operate on.

**Clustering trajectory views.** CTRL groups trajectories from the epoch's batch by clustering the set of their views $\{\boldsymbol{u}_i^{(\tau_i)}\}_{i=1}^{B}$. Since we would like to evolve clustering online as new trajectory batches arrive, we employ a common online clustering algorithm, the Sinkhorn-Knopp procedure (Caron et al., 2020), which has been used as an auxiliary RL loss (e.g., Proto-RL, Yarats et al., 2021). Since CTRL operates online, Sinkhorn-Knopp is better-suited for the task than other clustering methods.

The clustering branch computes two views of each input $\boldsymbol{u}_i^{(\tau_i)}$ in a cascading fashion: first by passing it through a clustering encoder $\psi_{\text{clust}} : \mathcal{U} \to \mathcal{V}$, e.g. an RNN, to obtain a lower-dimensional view $\boldsymbol{v}_i = \psi_{\text{clust}}(\boldsymbol{u}_i^{(\tau_i)})$, and then by passing $\boldsymbol{v}_i$ through yet another network, an MLP $\theta_{\text{clust}} : \mathcal{V} \to \mathcal{W}$, to produce view $\boldsymbol{w}_i = \theta_{\text{clust}}(\boldsymbol{v}_i)$. The parameters of $\psi_{\text{clust}}$ and $\theta_{\text{clust}}$ are learned through the epochs jointly with $\phi$'s. Like $\boldsymbol{u}_i^{(\tau_i)}$, each $\boldsymbol{v}_i$ and $\boldsymbol{w}_i$ is a view of trajectory $i$; the approach then consists in projecting $\boldsymbol{v}_i$'s and $\boldsymbol{w}_i$'s onto centroids of $C$ clusters in two different ways and then computes a clustering loss that enforces consistency between $\boldsymbol{v}_i$'s and $\boldsymbol{w}_i$'s cluster projections.

Specifically, we represent the centroid of each cluster $c$ with a vector $\boldsymbol{e}_c \in \mathcal{V}$, which are stacked into a matrix $\mathbf{E}$. These vectors are additional parameters in the joint optimization problem CTRL solves. They can be regarded as views of $C$ typical behaviors around which the trajectories' views are regrouped. To project trajectory $i$'s $\boldsymbol{v}_i$ views onto behavioral centroids learned from previous trajectory batches, CTRL computes a vector of soft assignments of $\boldsymbol{v}_i$ to each centroid $\boldsymbol{e}_c$:

$$\mathbf{Q}_i = \underset{1 \leq c \leq C}{\text{Softmax}} \left( \frac{\boldsymbol{v}_i^\top \boldsymbol{e}_c}{||\boldsymbol{v}_i||_2 ||\boldsymbol{e}_c||_2} \right) \tag{2}$$

---

[2]While, in theory, the Procgen suite is indeed a POMDP, most RL algorithms take the most recent observation as the belief state – a simplification which was shown not to hinder ZSG on Procgen (Cobbe et al., 2020).

and forms a $B \times C$ matrix $\mathbf{Q}$ whose $i$-th row is the soft assignment of $\boldsymbol{v}_i$. The resulting assignments may be very unbalanced, with most probability mass assigned to only a few clusters. Applying the Sinkhorn-Knopp algorithm solves this issue by iteratively re-normalizing $\mathbf{Q}$ in order to obtain a more equal cluster membership (Cuturi, 2013), where the degree of re-normalization is controlled by a temperature parameter $\beta$. The output of this operation is a matrix $\tilde{\mathbf{Q}}$.

For each view $\boldsymbol{w}_i$ (the projection of trajectory view $\boldsymbol{v}_i$) CTRL computes the logarithm of its soft cluster assignments and treats these vectors as rows of another $B \times C$ matrix $\mathbf{P}$:

$$\mathbf{P}_i = \log \left[ \underset{1 \leq c \leq C}{\text{Softmax}} \left( \boldsymbol{w}_i^\top \boldsymbol{e}_c \right) \right]. \tag{3}$$

Finally, we compute the cross entropy between $\tilde{\mathbf{Q}}$ and $\mathbf{P}$, which measures their inconsistency. This measure is taken as the clustering loss:

$$\mathcal{L}_{\text{clust}} = \text{CrossEntropy}(\tilde{\mathbf{Q}}, \mathbf{P}). \tag{4}$$

**Encouraging cross-cluster behavioral similarity.** Note, however, that the clustering loss in the above step emphasizes the recognition of behavioral similarities *within* clusters. This may hurt generalization, as the resulting centroids may not faithfully represent behaviors encountered at test time. Our hypothesis is that encouraging encoder $\phi$ to induce latent-space similarities between trajectories from *different but adjacent* clusters will increase its ability to recognize behaviors in unseen test trajectories.

While there are several ways to encourage cross-cluster representational similarity, using a mechanism similar to MYOW (Azabou et al., 2021) on trajectories drawn from neighboring clusters captures this idea particularly well. Namely, to get the cross-predictive loss $\mathcal{L}_{\text{pred}}$, we sample trajectory view pairs from neighboring clusters and apply the cosine-similarity loss to those pairs.

To implement this idea, we define a measure of cluster proximity via a matrix $\mathbf{D}$ of cosine similarities between cluster centroids: for clusters $k$ and $l$, $\mathbf{D}_{kl} = ||\boldsymbol{e}_k - \boldsymbol{e}_l||_2^2$ (Grill et al., 2020). Recall that in the previous step, the clustering branch computed a matrix $\mathbf{Q}$ whose rows $\mathbf{Q}_i$ are soft assignments of trajectory $i$'s view $\boldsymbol{v}_i$ to clusters. In this step, we convert these soft assignments to hard ones by associating a trajectory's view $\boldsymbol{v}_i$ with cluster $c_i = \text{argmax}_{1 \leq c' \leq C} \tilde{\mathbf{Q}}_i$ and treating a cluster $c$ as consisting of trajectories with indices in the set $\mathbb{T}_c = \{i \mid c = \text{argmax}_{1 \leq c' \leq C} \tilde{\mathbf{Q}}_i\}$. To assess how predictive a trajectory embedding $\boldsymbol{u}_i$ is of a trajectory embedding $\boldsymbol{u}_j$, like in the clustering step we will use two special helper maps, $\psi_{\text{pred}} : \mathcal{U} \to \mathcal{V}$ to obtain a reduced-dimensionality view $\boldsymbol{v}' = \psi_{\text{pred}}(\boldsymbol{u})$ and $\theta_{\text{pred}}$ to further project $\boldsymbol{v}'$ to $\boldsymbol{w}' = \theta_{\text{pred}}(\boldsymbol{v}')$.

CTRL proceeds by repeatedly sampling trajectories, which we call *anchor trajectories*, from the batch, with their associated embeddings $\boldsymbol{u}$. For each anchor trajectory $n$, consider $K$ clusters $c_1, \ldots, c_K$ nearest to $n$'s cluster $c_n$, as defined by the indices of $K$ largest values in row $c_n$ of matrix $\mathbf{D}$ (we exclude $c_n$ itself when determining $c_n$'s nearest clusters). Borrowing ideas from the MYOW approach Azabou et al. (2021), CTRL mines a view for $\boldsymbol{u}_n$ by randomly choosing a trajectory with embedding $\boldsymbol{u}_{c_k}^{(n)}$ from each of the neighboring clusters and computing its view $\boldsymbol{v}_{c_k}' = \psi_{\text{pred}}(\boldsymbol{u}_{c_k}^{(n)})$. We call the neighbors' views $\boldsymbol{v}_{c_1}', \ldots, \boldsymbol{v}_{c_K}'$ trajectory $n$'s *mined views*.

For the final operation in this step, CTRL computes trajectory $n$'s predictive view $\boldsymbol{w}_n' = \theta_{\text{pred}}(\psi_{\text{pred}}(\boldsymbol{u}_n))$ and measures the distance from it to trajectory $n$'s mined views:

$$\mathcal{L}_{\text{pred}}^{(n)} = \sum_{k=1}^{K} ||\boldsymbol{w}_n' - \boldsymbol{v}_{c_k}'||_2^2 \tag{5}$$

$N$ regulates the number of anchor trajectories to be sampled, so the total prediction loss is

$$\mathcal{L}_{\text{pred}} = \sum_{n=1}^{N} \mathcal{L}_{\text{pred}}^{(n)} \tag{6}$$

**Updating encoder $\phi$ using reward guidance without reward signal for representation learning.** Note that CTRL's total loss $\mathcal{L}_{\text{CTRL}}$ depends on the parameters of encoder $\phi$ as well as of clustering

networks $\phi_{\text{clust}}$ and $\theta_{\text{clust}}$, prediction networks $\phi_{\text{clust}}$ and $\theta_{\text{clust}}$, and cluster centroids $e_c$, $1 \leq c \leq C$. In each epoch, CTRL updates all these parameters to minimize $\mathcal{L}_{\text{CTRL}}$.

## 5 CONNECTION TO BISIMULATION

Deep bisimulation metrics are tightly connected to the underlying mechanism of mining behaviorally similar trajectories of CTRL. They operate on a latent-dimensional space and, as is the case for DeepMDP (Gelada et al., 2019) and DBC (Zhang et al., 2021), ensure that bisimilar states (i.e. behaviorally similar states with identical reward) are located close to each other in that latent space. In this section, we aim to highlight a functional similarity between bisimulation metrics and CTRL.

**Definition 1** *A bisimilation relation $E \subseteq \mathcal{S} \times \mathcal{S}$ is a binary relation which satisfies, $\forall (s, t) \in E$:*

1. *$\forall a \in \mathcal{A}, \mathcal{R}(s, a) = \mathcal{R}(t, a)$*

2. *$\forall a \in \mathcal{A}, \forall c \in \mathcal{S}, \sum_{s' \in c} \mathcal{P}(s, a)(s') = \sum_{s' \in c} \mathcal{P}(t, a)(s')$*

In practice, rewards and transition probabilities rarely match exactly. For this reason, Ferns et al. (2004) proposed a smooth alternative to bisimulation relations in the form of bisimulation metrics, which can be found by solving a recursive equation involving the Wasserstein-1 distance $\mathcal{W}_1$ between transition probabilities. $W_1$ can be found by solving the following linear programming (Villani, 2008), where we let $\Gamma = \{ \boldsymbol{v} \in \mathbb{R}^{|\mathcal{V}|} : 0 \leq \boldsymbol{v}_i \leq 1 \ \forall 1 \leq i \leq |\mathcal{V}| \}$:

$$\mathcal{W}_1^d(P||Q) := \max_{\mu \in \Gamma} \sum_{s \in \mathcal{S}} (P(s) - Q(s))\mu(s) \quad \text{s.t. } \mu(s) - \mu(s') < d(s, s') \forall s, s' \in \mathcal{S}, \quad (7)$$

where $\mu$ is a vector whose elements are constrained between 0 and 1. In practice, bisimulation metrics are used to enforce a temporal continuity of the latent space by minimization of the $\mathcal{W}_1$ loss between training state-action pairs. Therefore, to show a connection of CTRL to (reward-free) bisimulation metrics, it is sufficient to show that two trajectories are mapped to the same partition *if* their induced $\mathcal{W}_1$ distance is arbitrarily small. In our (informal) argument that follows, we assume that CTRL samples two consecutive timesteps and encodes them into $\boldsymbol{v}$; the exact form of $\boldsymbol{v}$ dictates the nature of the behavioral similarity. The proof can be found in Appendix 8.7.

**Proposition 1** *(Informal) Let $M$ be an MDP where $\mathcal{R}(s, a) = 0$ for all $(s, a) \in \mathcal{S} \times \mathcal{A}$ and let $\boldsymbol{v}, \boldsymbol{v}' \in \mathcal{V}$ be two dynamics embeddings in $M$. The clustering operation between $\boldsymbol{v}, \boldsymbol{v}'$ induces a reward-free bisimilarity metric $\mathcal{W}_1(\mathbb{P}[\boldsymbol{v}], \mathbb{P}[\boldsymbol{v}'])$ between induced distributions $\mathbb{P}[\boldsymbol{v}]$ and $\mathbb{P}[\boldsymbol{v}']$.*

## 6 EMPIRICAL EVALUATION

We compare CTRL against strong RL baselines: DAAC (Raileanu and Fergus, 2021) – the current state-of-the-art on the challenging generalization benchmark suite *Procgen* (Cobbe et al., 2020), and PPO (Schulman et al., 2017). DAAC optimizes the PPO loss (Schulman et al., 2017) through decoupling the training of the policy and value functions, which updates the advantage function during the policy network updates. We then compare to several unsupervised and SSL auxiliary objectives used in conjunction with PPO. DIAYN (Eysenbach et al., 2019) is an unsupervised skill-based exploration method which we adapt to the online setting by uniformly sampling skills. Its notion of skills has some similarities to the notion of clusters in CTRL. We also compare with two SSL-based auxiliary objectives: CURL (Srinivas et al., 2020), a common SSL baseline which contrasts augmented instances of the same state, and Proto-RL (Yarats et al., 2021), which we adapt for this generalization setting. Finally, we provide a comparison against bisimulation-based algorithms: DBC (Zhang et al., 2021), which was shown to perform well on robotic control tasks with visual distractor features, and PSE (Agarwal et al., 2021). PSE assumes policies for training tasks to be given and, like Agarwal et al. (2021), we ran it both with random and high-quality policies pretrained with extra computation budget. See Appendix 8.2 for details.

The Procgen benchmark suite, which we use in our experiments, consists of 16 video games (see Table 1). Procgen procedurally generates distinct levels for each game. The number of levels for each game is virtually unlimited. Levels within a game which share common game rules and objectives

but differ in level design such as the number of projectiles, background colors, item placements throughout the level and other game assets. All of this makes Procgen a suitable benchmark for zero-shot generalization. Using our notation from Section 3, for each of 16 games, we train on a uniform distribution $d(\mathcal{T}_N)$ over $N = 200$ "easy" levels of the game and evaluate on $d(\mathcal{T} \setminus \mathcal{T}_N)$, i.e., a uniform distribution over the game's "easy" levels not seen during training. Following Mohanty et al. (2021), we report results after 8M steps of training, since this demonstrates the quality of ZSG that various representation learning methods can achieve *quickly*. However, Figure 4 in Appendix 8.3 also provides results after 25M steps of training, as in the original Procgen paper (Cobbe et al., 2020).

| | RL | | RL+Bisim. | | | RL+Unsup. | RL+SSL | | Ours |
|---|---|---|---|---|---|---|---|---|---|
| Env | PPO | DAAC | PPO+DBC | PPO+PSE (random) | PPO+PSE (pretrained) | PPO+DIAYN | Proto-RL | PPO+CURL | **CTRL** |
| bigfish | 2.3±0.1 | 4.3±0.3 | 1.8±0.1 | 2.3 ± 0.1 | 1.8 ± 0.2 | 2.2±0.1 | 2.4±0.1 | 2.2±0.2 | **4.7±0.2** |
| bossfight | 5.2±0.3 | 1.7±0.7 | 5±0.1 | 0.9 ± 0.2 | 0.7 ± 0.1 | 1.1±0.2 | 6.1±0.5 | 4.6±0.8 | **8.2±0.1** |
| caveflyer | 4.4±0.3 | 4.3±0.1 | 3.6±0.1 | 2.6 ± 0.1 | 3.6 ± 0.3 | 1.0±1.9 | **4.7±0.1** | 4.6±0.3 | **4.7±0.2** |
| chaser | 7.2±0.2 | 7.1±0.1 | 4.8±0.1 | **8.7 ± 0.5** | 4.2 ± 0.2 | 5.6±0.5 | 7.6±0.2 | 7.2±0.2 | 7.1±0.2 |
| climber | 5.1±0.1 | 5.5±0.2 | 4.1±0.4 | 2.9 ± 0.1 | 3.9 ± 0.2 | 0.8±0.7 | 5.5±0.3 | 5.5±0.1 | **5.9±0.2** |
| coinrun | 8.3±0.2 | 8.1±0.1 | 7.9±0.1 | 5.5 ± 0.5 | 7.3 ± 0.2 | 6.4±2.6 | 8.2±0.1 | 8.1±0.1 | **8.7±0.3** |
| dodgeball | 1.3±0.1 | **1.8±0.2** | 1.0±0.3 | 1.6 ± 0.1 | 1.3 ± 0.1 | 1.4±0.2 | 1.6±0.1 | 1.4±0.1 | **1.8±0.1** |
| fruitbot | 12.4±0.2 | 11.5±0.3 | 7.6±0.2 | 1.0 ± 0.1 | 1.1 ± 0.2 | 7.2±3.0 | 12.3±0.4 | 12.3±0.2 | **13.3±0.3** |
| heist | 2.7±0.2 | **3.4±0.2** | 3.3±0.2 | 3.1 ± 0.3 | 2.9 ± 0.4 | 0.2±0.2 | 3.0±0.3 | 2.5±0.1 | 3.1±0.3 |
| jumper | 5.8±0.3 | **6.3±0.1** | 3.9±0.4 | 4.1 ± 0.3 | 5.3 ± 0.2 | 2.6±2.3 | 6.0±0.1 | 5.9±0.1 | 6.0±0.1 |
| leaper | 3.5±0.4 | 3.5±0.4 | 2.7±0.1 | 2.7 ± 0.2 | 2.6 ± 0.1 | 2.5±0.2 | 3.2±0.8 | **3.6±0.5** | 2.8±0.2 |
| maze | 5.4±0.2 | 5.6±0.2 | 5.0±0.1 | 5.4 ± 0.2 | 5.3 ± 0.1 | 1.6±1.2 | 5.5±0.3 | 5.4±0.1 | **5.7±0.1** |
| miner | 8.7±0.3 | 5.7±0.1 | 4.8±0.1 | 5.6 ± 0.1 | 4.3 ± 0.3 | 1.3±2.0 | 8.8±0.5 | **8.6±0.2** | 6.5±0.2 |
| ninja | 5.5±0.2 | 5.2±0.1 | 3.5±0.1 | 3.4 ± 0.2 | 3.5 ± 0.3 | 2.8±2.0 | 5.4±0.3 | 5.6±0.1 | **5.8±0.1** |
| plunder | 6.2±0.4 | 4.1±0.1 | 5.1±0.1 | 4.0 ± 0.1 | 4.1 ± 0.2 | 2.1±2.5 | 6.0±0.9 | 6.5±0.3 | **6.6±0.3** |
| starpilot | 4.7±0.2 | 4.1±0.2 | 2.8±0.1 | 3.2 ± 0.2 | 3.0 ± 0.1 | 5.8±0.6 | 5.2±0.2 | 5.0±0.1 | **7.7±0.5** |

Table 1: Average evaluation returns collected after 8M training frames, $\pm$ one standard deviation over 10 seeds.

**Main results.** As Table 1 shows, PPO+CTRL outperforms all other baselines, including DAAC, on most games. Notably, bisimulation-based approaches other than CTRL– DBC as well as PSE with both random and expert data-gathering policies – exhibit lower gains than others. While this can be surprising, recent work has seen similar results when applying DBC to tasks with unseen backgrounds (Li et al., 2021). PSE's inferior performance may be due to the policy similarity metric, which PSE requires as input and which we took from Agarwal et al. (2021), being poorly suited to Procgen. This highlights an important difference between CTRL and PSE: CTRL doesn't need a policy similarity metric, since it implicitly induces such a metric based on trajectory "signatures".

Despite training static prototypes for 8M timesteps and adapting the RL head for 8M additional timesteps (see Appendix 8.2 for details), Proto-RL performs worse than CTRL. This suggests that the temporal aspect of clustering is key for ZSG, a hypothesis we explore further in Section 8.4. Likewise, PPO+DIAYN uses its pre-training phase to find a diverse set of skills, which can be useful in robotics domains, but does not help much in the ZSG setting of Procgen. DAAC also exhibits good generalization performance, but inherits from PPG (Cobbe et al., 2021) the separation of the value and policy branches both parameter-wise and by introducing distinct training phases, an overhead which CTRL manages to avoid. In addition, we describe a number of ablation studies (Appendix 8.3), empirically show that slow clustering convergence leads to better generalization (Appendix 8.4), and demonstrate on a toy example how learning behavioral similarities captures local perceptual changes (Appendix 8.5).

## 7 CONCLUSIONS

This work proposed CTRL, a novel representation learning algorithm that facilitates zero-shot generalization of RL policies in high-dimensional observation spaces. CTRL can be viewed as inducing an unsupervised reward-agnostic bisimulation metric over observation histories, learned over transitions encountered by policies from an RL algorithm's *value improvement path* (Dabney et al., 2021). We hope that in the future CTRL will inspire other representation learning methods based on capturing belief states's behavioral similarity, which will be capable of policy generalization across greater variations in environment dynamics.

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

# 8 APPENDIX

## 8.1 CTRL PSEUDOCODE

---

**Algorithm 1:** Cross Trajectory Representation Learning

---

**Inputs** : online encoder $\phi$, cluster projector $\theta_{\text{clust}}$, cluster encoder $\psi_{\text{clust}}$, mining projector $\theta_{\text{pred}}$, mining encoder $\psi_{\text{pred}}$ , cluster basis matrix $\mathbf{E}$

**Hyperparameters :** $B$ – trajectory batch size, $C$ – num. of trajectory clusters, $T$ – subtrajectory length, $K$ – num. of nearest clusters for view mining, $L$ – trajectory length, $N$ – num. of anchors for view mining, $\beta$ – Sinkhorn temperature

1 **for** *each iteration $itr = 1, 2, ..$* **do**
2     **for** *each minibatch $\mathcal{B}$* **do**
3        **for** *each trajectory $\tau_i$ in $\mathcal{B}$* **do**
4           $t_1, .., t_T \sim$ Uniform($L$) `// Sample temporal keypoints`
5           $u^{(\tau_i)} = [FiLM(\phi(s_{t_1}), a_{t_1}), ..., FiLM(\phi(s_{t_T}), a_{t_T})]$
       `// cluster dynamics`
6        $\boldsymbol{u}_i = [u^{(\tau_1)}, ..., u^{(\tau_m)}]$ `// batch dynamics`
7        $\boldsymbol{v}_i = \psi_{\text{clust}}(\boldsymbol{u}_i)$ `// fetch embeddings`
8        $\boldsymbol{w}_i = \theta_{\text{clust}}(\boldsymbol{v}_i)$ `// fetch projections`
9        $\boldsymbol{v}_i = \frac{\boldsymbol{v}_i}{||\boldsymbol{v}_i||_2}$ `// normalize embeddings`
10        $\mathbf{Q} = \text{Softmax}\big(\boldsymbol{v}_i^\top \mathbf{E}/\beta\big)$ `// Compute latent dynamics scores`
11        $\tilde{\mathbf{Q}} = \text{Sinkhorn}(\mathbf{Q})$ `// normalize scores through Sinkhorn`
12        $\mathbf{P} = \log\big[\text{Softmax}(\boldsymbol{w}_i^\top \mathbf{E}/\beta)\big]$ `// Compute projected dynamics scores`
13        $\mathcal{L}_{\text{clust}}(\phi, \psi_{\text{cluster}}, \theta_{\text{cluster}}) = \text{CrossEntropy}(\tilde{\mathbf{Q}}, \mathbf{P})$
       `// Predicting neighbors`
14        $\mathbf{D}_{ij} = ||\boldsymbol{e}_i - \boldsymbol{e}_j||_2^2$ `// find pairwise basis distances`
15        $\mathcal{L}_{\text{pred}} = 0$
16        **for** *each anchor $j = 1, ..., N$* **do**
17           $\tau_j \sim \mathcal{B}$ `// Sample anchor trajectory`
18           $\boldsymbol{u}_n = u^{(\tau_j)}$ `// Set anchor embedding`
19           $c_i^{(1)}, .., c_i^{(k)} = \text{top-knn}(\mathbf{D}, k, c_i)$ `// Find nearby clusters`
20           $u_{c_1}, ..., u_{c_k} \sim p(u_{c_i^{(1)}}), ..., p(u_{c_i^{(k)}})$ `// Sample views from clusters`
21           $\boldsymbol{v}'_{c_1}, ..., \boldsymbol{v}'_{c_K} = \psi_{\text{pred}}(u_{c_1}), ..., \psi_{\text{pred}}(u_{c_k})$ `// embed mined views`
22
23           $\boldsymbol{w}'_n = \theta_{\text{pred}}(\psi_{\text{pred}}(\boldsymbol{u}_n))$ `// mining target`
24           $\mathcal{L}_{\text{pred}}^{(n)} = \sum_{k=1}^K ||\boldsymbol{w}'_n - \text{StopGrad}(v'_{c_k})||_2^2$
25        $\mathcal{L}_{\text{pred}} = \sum_{n=1}^N \mathcal{L}_{\text{pred}}^{(n)}$
26        $\mathcal{L}_{\text{CTRL}} = \mathcal{L}_{\text{clust}} + \mathcal{L}_{\text{pred}}$ `// update networks`
27        $\phi, \psi_{\text{clust}}, \theta_{\text{clust}}, \theta_{\text{pred}}, \psi_{\text{pred}} = \text{Adam}(\phi, \psi_{\text{clust}}, \theta_{\text{clust}}, \theta_{\text{pred}}, \psi_{\text{pred}}; \mathcal{L}_{\text{CTRL}})$
28     **for** *each minibatch $\mathcal{B}$* **do**
29        $\pi, V^\pi = \text{Adam}(\pi, V^\pi; \mathcal{L}_{\text{RL}}(\mathcal{B}))$ `// update RL parameters`

---

## 8.2 EXPERIMENT DETAILS

We implemented all algorithms on top of the IMPALA architecture, which was shown to perform well on Procgen (Cobbe et al., 2020). In the Procgen experiments, **Proto-RL** was ran without any intrinsic rewards (since the domains are not exploration-focused) by first jointly training the representation and RL losses for 8M timesteps, after which only the RL loss was optimized for an additional 8M steps (16M steps in total). Similarly, **DIAYN** was also run with the pre-training phase of 8M and then RL only objective for the second 8M phase.

Like Proto-RL and DIAYN, **PSE** needed extra training budget and extra adjustments for a fair comparison to CTRL (see Section 2). Agarwal et al. (2021) used PSE only with Soft Actor-Critic, which doesn't perform well on Procgen. Therefore, we carried over PSE' available implementation

| Name | Description | Value |
|---|---|---|
| $\gamma$ | Discount factor | 0.999 |
| $\lambda$ | Decay | 0.95 |
| $n_{\text{timesteps}}$ | Number of timesteps per rollout | 256 |
| $n_{\text{epochs}}$ | Number of epochs for RL and representation learning | 1 |
| $n_{\text{samples}}$ | Number of samples per epoch | 8192 |
| Entropy bonus | Entropy loss coefficient | 0.01 |
| Clip range | Clip range for PPO | 0.2 |
| Learning rate | Learning rate for RL and representation learning | $5 \times 10^{-4}$ |
| Number of environments | Number of parallel environments | 32 |
| Optimizer | Optimizer for RL and representation learning | Adam |
| Frame stack | Frame stack $X$ Procgen frames | 1 |
| $E$ | Number of clusters | 200 |
| $k$ | Number of k-NN nearest neighbors | 3 |
| $T$ | Number of clustering timesteps | 2 |
| $\beta$ | Clustering temperature | 0.3 |

Table 2: Experiments' parameters

from https://agarwl.github.io/pse/ to our codebase, with the help of PSE' authors, to combine it with the same PPO implementation that CTRL used.

PSE assumes being given policies for training problem instances (Procgen levels). Like Agarwal et al. (2021), we ran PSE using both random and high-quality pretrained policies for these problems. In the latter case, we pretrained expert policies for the first 40 levels of Procgen to generate training trajectories for PSE. Each level's expert was trained on 0.5M environment steps. Note that this is much less than 8M steps we used for policy training in experiments with other algorithms, but this is because each expert needed to be good *only for a single level*, and we verified that they indeed were. We did this only for the first 40 levels because even for 40 levels this took 20M training steps *per game* and had to be done for 16 games. We don't believe more than 40 experts per game would have made a difference.

Given policies for the training levels, PPO+PSE's training on Procgen mimicked CTRL's: PPO+PSE trained by interacting with the first 200 levels for 8M steps and was evaluated on the rest. However, during the training, PPO+PSE sampled an additional 1M interactions from the pretrained traiing-level policies. the process was repeated for all 16 games, for 10 seeds each.

We did hyperparameter grid search on PPO+PSE's hyperparameters for PSE – loss coefficient values of $(0.1, 1, 2)$ and temperature $(0.1, 0.3, 0.7)$. PPO's hyperparameters were the same as in CTRL.

Thus, due to the need to pretrain and gather data with per-level expert policies, PPO+PSE received $0.5 \cdot 20M + 1M = 21M$ extra environment interactions compared to CTRL, i.e., used $21M/8M = 2.6\times$ more training data than the latter.

### 8.3 ADDITIONAL RESULTS

**ZSG over 8M and 25M training steps and policy performance throughout training**  We provide the performance plots of various representation learning and RL algorithms for the 8M and the 25M benchmarks, which test zero-shot generalization under different sample regimes. Figure 2 shows the training performance of agents on 8M frames, Figure 3 the test performance of agents on 8M frames and all levels, and, finally, Figure 4 shows the test performance of agents on 25M frames and all levels.

**Ablations on algorithm components**  We ran multiple versions of the algorithm to identify the key components which make CTRL perform well in Procgen. The first modification, **CTRL consecutive T** consists in running our algorithm but sampling consecutive timesteps, that is $t_{i+1} = t_1 + i$ for all $t_1$ and $0 \le i \le T$. The second modification, **CTRL no action** removes the action conditioning layer in the log-softmax probability $p_t$ and in the cluster scores $q_t$, to test the importance of action information for cluster membership prediction. The third modification, **CTRL no cluster** removes the clustering

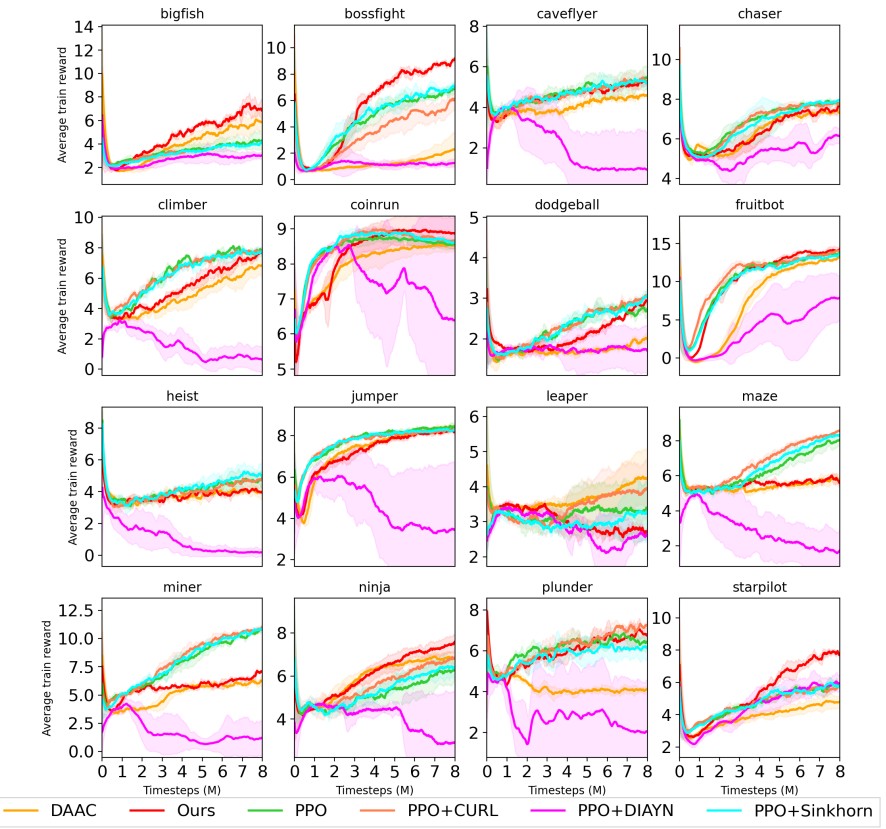

Figure 2: Training results over the 8M frames benchmark.

loss, and only restricts to mining and predicting nearby neighbors in the batch. Finally, the last modification, **CTRL no pred** removes the loss predicting samples from neighboring partitions and only relies on the clustering loss to update its representation.

Results suggest that (1) using consecutive timesteps as for the dynamics vector embedding yields lower average rewards than non-consecutive timesteps, (2) action conditioning helps the agent to pick up on the local dynamics present in the MDP and (3) both clustering and predictive objectives are essential to the good performance of our algorithm. Results of the last column are computed over 10 seeds, rest over 3 seeds.

**Ablation on number of clusters and clustering timesteps**    How should one determine the optimal number of clusters in a complex domain? Can the number of clusters be chosen *a priori* running any training?

Below, we provide some partial answers to these questions. First, the optimal (or true) number of clusters is domain-specific, as it depends on the exact connectivity structure of the MDP at hand. Second, the length of the clusters, i.e. the number of trajectory timesteps passed to Sinkhorn-Knopp can widely impact the nature of learned representations, and hence the downstream performance of the agent.

**Temporal connectivity of the clusters**    Are clusters *consistent* in time for a given trajectory? To verify this, we trained CTRL  on 1 million frames of the *bigfish* game. For every $T$ states in a given trajectory, we have computed the hard cluster assignment to the nearest cluster, which yields

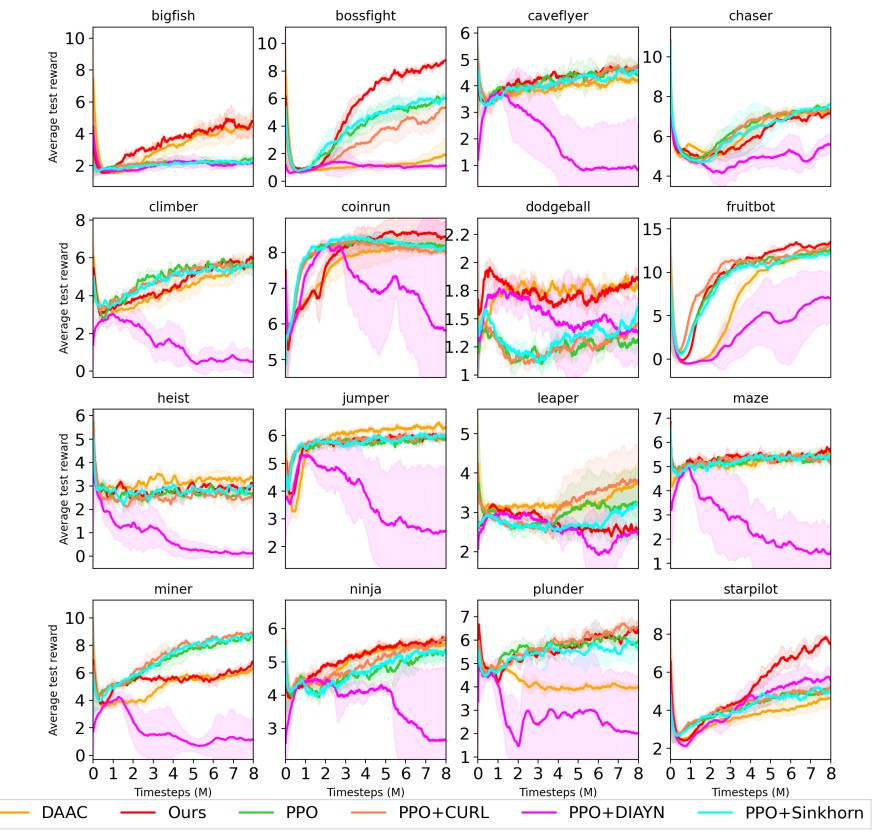

Figure 3: Evaluation results over the 8M frames benchmark.

Table 3: Average evaluation returns collected after 8M of training frames, ± one standard deviation.

| Env | CTRL consecutive T | CTRL no action | CTRL no cluster | CTRL no pred | CTRL |
|---|---|---|---|---|---|
| bigfish | 3.9±0.3 | 3.2±0.3 | 2.5±0.3 | 3.7±0.1 | 4.7±0.2 |
| bossfight | 8.9±0.1 | 6.9±0.9 | 7.8±0.3 | 6.6±0.8 | 8.2±0.1 |
| caveflyer | 4.6±0.2 | 4.7±0.1 | 4.6±0.1 | 4.6±0.1 | 4.7±0.2 |
| chaser | 7.4±0.3 | 6.7±0.2 | 7.0±0.5 | 6.5±0.1 | 7.1±0.2 |
| climber | 6.2±0.4 | 5.5±0.1 | 5.3±0.4 | 5.7±0.4 | 5.9±0.2 |
| coinrun | 8.8±0.1 | 8.5±0.3 | 8.1±0.2 | 8.4±0.3 | 8.7±0.3 |
| dodgeball | 1.8±0.1 | 1.7±0.1 | 1.7±0.2 | 1.7±0.1 | 1.8±0.1 |
| fruitbot | 13.1±0.3 | 12.9±0.4 | 13.0±0.5 | 12.5±0.6 | 13.3±0.3 |
| heist | 3.0±0.1 | 3.2±0.3 | 3.2±0.1 | 3.0±0.2 | 3.1±0.3 |
| jumper | 6.1±0.2 | 6.0±0.1 | 5.9±0.1 | 5.9±0.1 | 6.0±0.1 |
| leaper | 3.4±1.1 | 3.2±0.4 | 2.6±0.3 | 3.3±0.2 | 2.8±0.2 |
| maze | 5.6±0.2 | 5.6±0.1 | 5.7±0.1 | 5.8±0.1 | 5.7±0.1 |
| miner | 7.0±0.9 | 5.9±0.4 | 5.6±0.1 | 6.0±0.2 | 6.5±0.2 |
| ninja | 5.7±0.1 | 5.3±0.2 | 5.5±0.1 | 5.6±0.1 | 5.8±0.1 |
| plunder | 6.4±0.2 | 5.6±0.1 | 5.9±0.5 | 6.1±0.2 | 6.6±0.3 |
| starpilot | 7.0±0.2 | 4.9±0.6 | 4.9±0.2 | 5.8±0.4 | 7.7±0.5 |

a sequence of partitions. We then computed the cosine similarity between time-adjacent cluster centroids, the metric reported on the smoothed graph below.

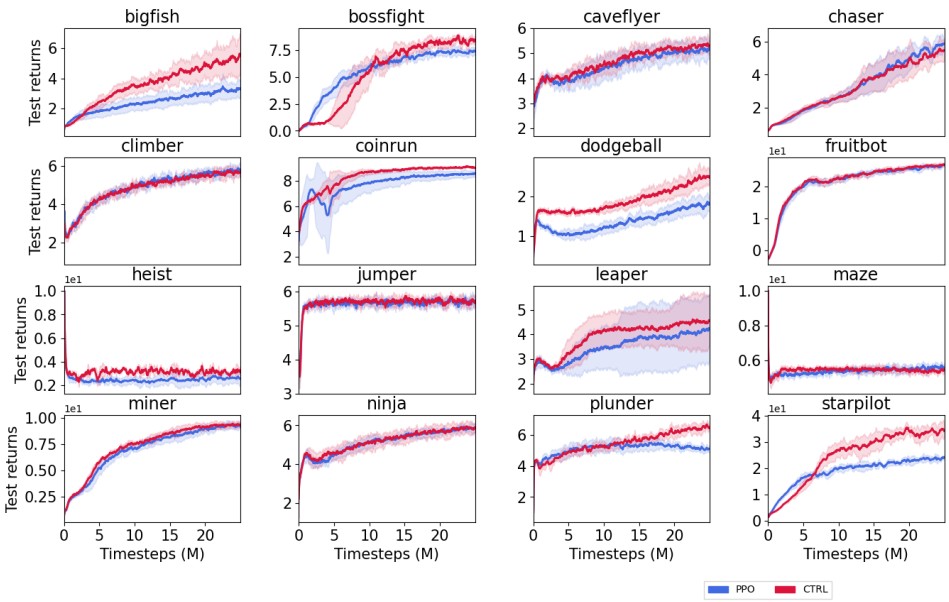

Figure 4: Evaluation results over the 25M frames benchmark.

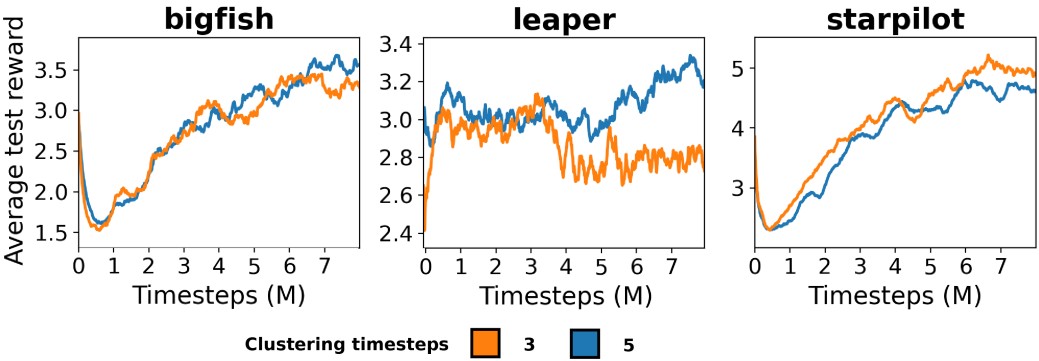

Figure 5: Ablation on the clustering timesteps used in the dynamics embedding

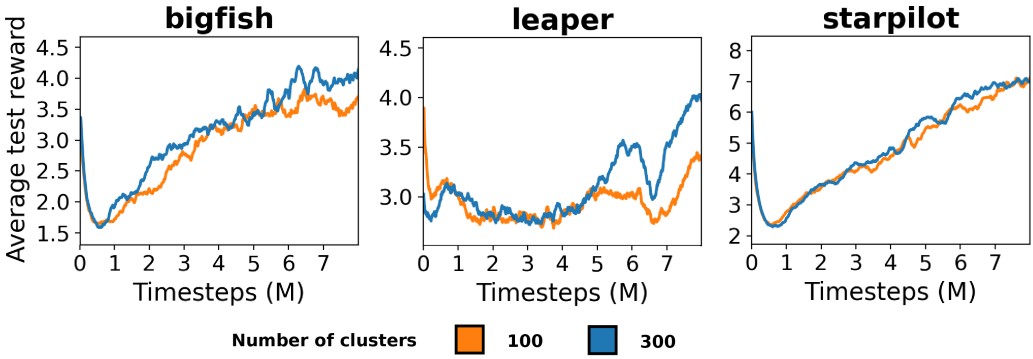

Figure 6: Ablation on the number of clusters used in CTRL

**Loss landscape of $\mathcal{L}_{\text{clust}}$ and $\mathcal{L}_{\text{pred}}$**    Works relying on non-colinear signals, e.g. behavioral similarity and rewards, as is the case for DeepMDP (Gelada et al., 2019), show that interference can occur between various loss components. For example, (Gelada et al., 2019) showed how their dynamics and

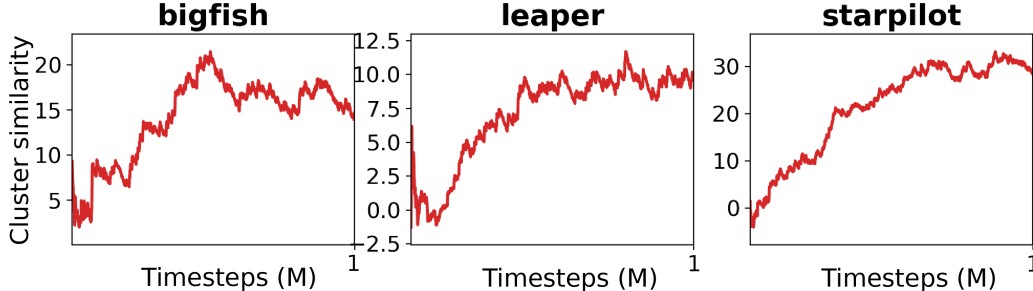

Figure 7: Average within-trajectory cluster similarity over 1M consecutive timesteps.

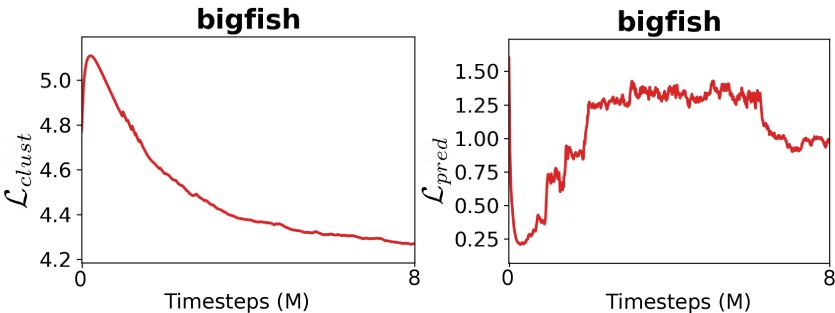

Figure 8: Average values of $\mathcal{L}_{clust}$ and $\mathcal{L}_{pred}$ over time.

reward losses are inversely proportional to each other early on in the training, taking a considerable amount of frames to converge.

We observe a similar pattern in Figure 8: the clustering loss first jumps up while the predictive loss is minimized, then the trend reverses, and both losses get minimized near the end of the training.

**Case study: splitting other dynamics-aware losses** Similar to the postulate of ATC (Stooke et al., 2021), we hypothesize that training the encoder only with the representation loss has the most beneficial effect when the representation loss contains information about dynamics. To validate this, we conducted an additional set of experiments on two well-known self-supervised learning algorithms which leverage predictive information about future timesteps: Deep Reinforcement and InfoMax Learning (DRIML) (Mazoure et al., 2020) and Self-Predictive Representations (SPR) (Schwarzer et al., 2021). We ran (i) the default version of the algorithms with joint RL and representation updates, as well as (ii) RL updates propagated only through the layers above the encoder.

**Qualitative assessment of clusters** Figure 9 shows, for 5 environments, 4 randomly sampled states for 2 behavioral clusters (4 clusters for Starpilot – differences between clusters are easier to visualize in this environment). Note that clustered states go beyond visual similarity, and capture action sequences, agent position, presence of enemies and even topological equivalence of various levels. The choice of environments for this demonstration is dictated by the nature of the action space, e.g. projectiles in Starpilot and path tracing in Miner allow to better visualize agent's behavior. Note that, for Bigfish, CTRL implicitly picks up the notion of reward density by learning to separate states abundant of fish from those without fish (due to the policy being trained on rewards and thus exhibiting different behavior in those two settings).

Figure 10 shows the t-SNE of from randomly sampled states along the CTRL training path on Starpilot – embeddings learned by CTRL can be seen to concentrate into distinct clusters and around their respective centroids.

| Env | DRIML | SPR |
| --- | --- | --- |
| bigfish | **+0.17** | **+0.15** |
| bossfight | **+0.56** | **+10.36** |
| caveflyer | **+0.23** | -0.02 |
| chaser | -0.14 | -0.13 |
| climber | **0.15** | **+0.14** |
| coinrun | -0.37 | **+2.05** |
| dodgeball | -0.39 | **+0.12** |
| fruitbot | -0.53 | -0.1 |
| heist | -0.3 | -0.03 |
| jumper | **+0.04** | +0 |
| leaper | -0.04 | **+0.08** |
| maze | **+0.01** | -0.15 |
| miner | **+0.88** | **+0.03** |
| ninja | -0.13 | -0.31 |
| plunder | -0.02 | -0.18 |
| starpilot | -0.11 | -0.15 |
| Norm. score | +0.01 | +0.74 |

Table 4: Normalized improvement scores of split updates over joint updates of the encoder, averaged over 3 random seeds.

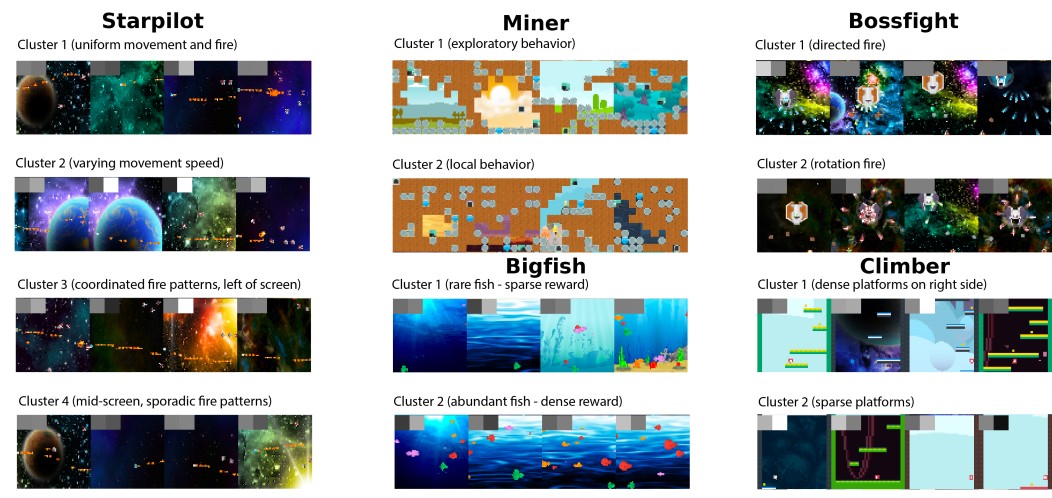

Figure 9: Sample states from behavioral clusters found by CTRL after 2M of training frames for 5 representative environments. The two gray squares in top left is added to indicate the agent's velocity.

## 8.4 SHOWCASE: SLOW CLUSTERING CONVERGENCE LEADS TO BETTER GENERALIZATION

Trajectory clustering is key to representation learning not only in CTRL but also in a prior method, Proto-RL. However, while Proto-RL uses it to pretrain a representation which it then keeps frozen during RL, CTRL applies clustering to evolve the representation as RL progresses. This raises a question: how important is online clustering convergence rate for learning a good representation? Intuitively, if online clustering converges too quickly and behavioral similarities are "pinned down" early in the training process, the resulting representation will not be robust to distribution shifts induced by improved policies. Therefore, it seems crucial to learn the behavioral similarities at a rate that allows cluster centroids to adapt to the value improvement path (Dabney et al., 2021). To validate this hypothesis, we conducted an analysis of the correlation between the clustering quality and the test performance as a function of training progress on three representative games.

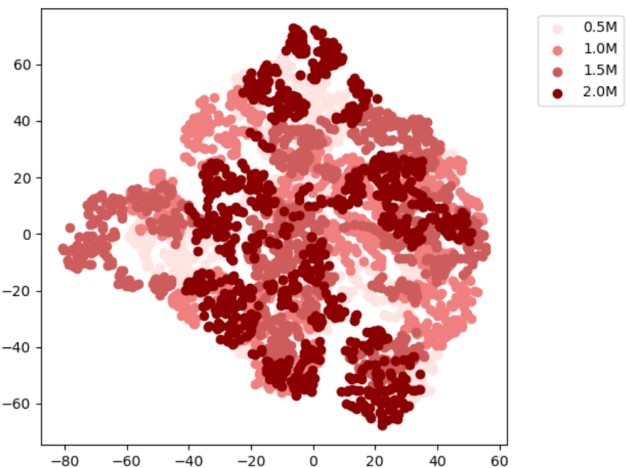

Figure 10: t-SNE of 1024 randomly sampled states from data collected by CTRL after 0.5,1,1.5 and 2M frames in Starpilot, with $\beta = 0.1, T = 4$. As learning progresses, agent behavior clusters become more and more distinct.

To measure the clustering quality, we report the silhouette score (Rousseeuw, 1987), a commonly used unsupervised goodness-of-fit measure which balances inter- and intra-cluster variance. Results shown

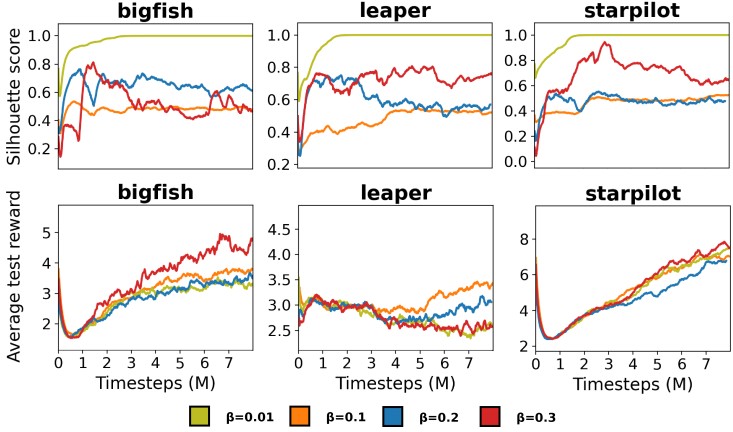

Figure 11: Goodness-of-clustering measured by silhouette scores (top) and average test returns (bottom) as a function of training samples.

in Figure 11 provide evidence that picking the unsupervised learning procedure which converges the fastest (i.e. uses the lowest temperature $\beta$) does not necessarily lead to the best generalization performance. Based on Figure 11, we conjecture that fast clustering convergence hinders the performance of the RL agent due to clusters being fixed early on and not adapting to the distribution shift induced by the evolving RL policy.

## 8.5 SHOWCASE: LEARNING BEHAVIORAL SIMILARITIES CAPTURES LOCAL PERCEPTUAL CHANGES

To demonstrate the importance of identifying behavioral similarities, we designed a toy example problem with 5 behavioral clusters, where clustering the behaviors correctly leads to finding a near-optimal policy.

Our example problem is based on the standard Ising model[3] – a $32 \times 32$ binary lattice, each entry of which evolves at every timestep according to the values of its neighbors, with strengths of neighbor dependencies being regulated by a temperature parameter $1/\beta$. We randomly initialize 5 Ising models, each parametrized by an inverse temperature parameter on a uniform grid $\beta \in [0.01, 0.3]$. The system state is given by the state of all 5 models, and all models evolve in parallel at every step. At each timestep, the agent needs to choose one of the models, and has 5 actions corresponding to these choices. At timestep $t$, the agent is allowed to observe only the state of the model it chose at this step, and gets a reward based only on this model's state. The reward yielded by Ising model $i$ at timestep $t$ is given by $r_t = -||s_{i,t} - G||_2^2$, where $G$ is a goal state. For a given problem instance, we sample $G$ randomly by instantiating a 6th Ising model with an unknown inverse temperature parameter $\beta^*$ and letting $G$ be its final configuration after evolving it for a random number of steps. Figure 12 outlines the experimental setting for this study case.

The 5 behavioral clusters in our setting correspond to the 5 Ising models. The optimal strategy to solve this task is to 1) identify the Ising model (i.e., the behavioral cluster) whose temperature parameter is closest to $\beta^*$ and 2) choose that model and collect the corresponding rewards.

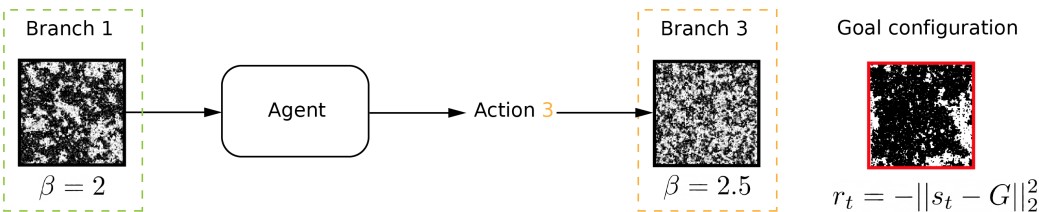

Figure 12: The composite Ising matching problem: the agent has to match a given Ising configuration by swapping branches of various transition dynamics

Table 5 outlines the results we obtained by deploying CTRL with different number-of-clusters parameter value. One can see that the largest improvement in silhouette score occurs from $E = 4$ to $E = 5$ (14.5%), suggesting that monitoring the largest change in silhouette score can be used to set the true number of clusters in CTRL which, in turn, corresponds to the highest-return policy discovered by CTRL.

|  | $E = 2$ | $E = 4$ | $E = 5$ | $E = 6$ | ... | $E = 50$ |
|---|---|---|---|---|---|---|
| Returns | -0.78 | -0.96 | -0.02 | -0.07 | ... | -0.54 |
| Silhouette | 0.875 | 0.796 | 0.651 | 0.554 | ... | 0.039 |
| Silhouette change | - | 0.079 | **0.145** | 0.097 | ... | 0.515 |

Table 5: Returns and silhouette scores obtained by CTRL in the composite Ising matching domain.

## 8.6 ADDITIONAL THEORETICAL FINDINGS

**Do uncorrelated local changes to state embeddings affect the clustering?**

**Theorem 8.1** *Let $M$ be an MDP and let $\boldsymbol{v} \in \mathcal{V}$ be a dynamics embeddings in $M$. Define*

$$\begin{cases} \boldsymbol{\delta}_i = \boldsymbol{\delta}_i^1 & 1 \leq i \leq |\mathcal{V}| \\ \boldsymbol{\delta}_i = \boldsymbol{\delta}_i^2 & h < i \leq T|\mathcal{V}| \end{cases} \tag{8}$$

*and pick $\boldsymbol{\delta}^1$ s.t. it lies on the positive half-plane spanned by $\mathbf{E}_j^\top - \mathbf{E}_{j'}^\top$ for some $1 \leq j' \leq E$. Then, $\boldsymbol{v}' = \boldsymbol{v} + \boldsymbol{\delta}$ and $\boldsymbol{v}$ belong to the same partition $j$.*

It becomes apparent from the above statement that perturbations to a single state or groups of state embeddings *do not* modify the partition membership as long as their direction aligns with that of the cluster embeddings.

---

[3]https://en.wikipedia.org/wiki/Ising_model

## 8.7 PROOFS

Throughout this section, we assume that the policy $\pi$ is fixed, and that CTRL optimizes $\mathcal{L}_{clust}$ only.

**Proof 1 (Theorem 8.1)** *For two dynamics embeddings to be assigned to the same cluster $j$, the following should hold*

$$\sum_{i=1}^{|\mathcal{V}|} \boldsymbol{v}_i \mathbf{E}_{ji}^\top > \sum_{i=1}^{|\mathcal{V}|} \boldsymbol{v}_i \mathbf{E}_{j'i}^\top,$$

$$\sum_{i=1}^{|\mathcal{V}|} (\boldsymbol{v}_i + \boldsymbol{\delta}_i) \mathbf{E}_{ji}^\top > \sum_{i=1}^{|\mathcal{V}|} (\boldsymbol{v}_i + \boldsymbol{\delta}_i) \mathbf{E}_{j'i}^\top \tag{9}$$

*for any $1 \leq j' \leq E$ s.t. $j' \neq j$.*

$$\sum_{i=1}^{|\mathcal{V}|} \boldsymbol{v}_i \mathbf{E}_{ji}^\top + \sum_{i=1}^{|\mathcal{V}|} \boldsymbol{\delta}_i \mathbf{E}_{ji}^\top > \sum_{i=1}^{|\mathcal{V}|} \boldsymbol{v}_i \mathbf{E}_{j'i}^\top + \sum_{i=1}^{|\mathcal{V}|} \boldsymbol{\delta}_i \mathbf{E}_{j'i}^\top$$

$$\sum_{i=1}^{|\mathcal{V}|} \boldsymbol{v}_i (\mathbf{E}_{ji}^\top - \mathbf{E}_{j'i}^\top) + \sum_{i=1}^{|\mathcal{V}|} \boldsymbol{\delta}_i (\mathbf{E}_{ji}^\top - \mathbf{E}_{j'i}^\top) > 0 \tag{10}$$

*Taking the difference between both equations yields the necessary condition for two dynamics to belong to the same cluster*

$$\sup_{1 \leq j' \leq E} (\mathbf{E}_{ji}^\top - \mathbf{E}_{j'i}^\top) \boldsymbol{\delta}_i \geq 0, \ 1 \leq i \leq |\mathcal{V}|. \tag{11}$$

**Corollary 8.1.1** *Let $\boldsymbol{v}, \boldsymbol{v}'$ be two dynamics embeddings, and define $\boldsymbol{\delta} = \boldsymbol{v}' - \boldsymbol{v}$. If $\boldsymbol{v}$ belongs to cluster $j$ and $j = \arg\max_{1 \leq j' \leq E} \mathbf{E}_{j'}^\top \boldsymbol{\delta}$, then $\boldsymbol{v}'$ also belongs to cluster $j$.*

*Perturbations are of the form $\sum_{i=1}^{|\mathcal{V}|} (\mathbf{E}_{ij}^\top - \mathbf{E}_{ij'}^\top) \boldsymbol{\delta}_i$. If $\boldsymbol{\delta} = 0$, then the cluster assignment doesn't change. Let $\boldsymbol{v}$ be of size $kh = |\mathcal{V}|$. Define, without loss of generality*

$$\begin{cases} \boldsymbol{\delta}_i = \boldsymbol{\delta}_i^1 & 1 \leq i \leq h \\ \boldsymbol{\delta}_i = \boldsymbol{\delta}_i^2 & h < i \leq kh \end{cases} \tag{12}$$

*and pick $\boldsymbol{\delta}^1$ s.t. it lies on the positive half-plane spanned by $\mathbf{E}_{ij}^\top - \mathbf{E}_{ij'}^\top$.*

*Then,*

$$\sum_{i=1}^{|\mathcal{V}|} (\mathbf{E}_{ij}^\top - \mathbf{E}_{ij'}^\top) \boldsymbol{\delta}_i = \sum_{i=1}^{h} (\mathbf{E}_{ij}^\top - \mathbf{E}_{ij'}^\top) \boldsymbol{\delta}_i^1 + \sum_{i=h}^{kh} (\mathbf{E}_{ij}^\top - \mathbf{E}_{ij'}^\top) \boldsymbol{\delta}_i^2 \geq \sum_{i=h}^{kh} (\mathbf{E}_{ij}^\top - \mathbf{E}_{ij'}^\top) \boldsymbol{\delta}_i^2 \geq 0 \tag{13}$$

*which concludes the proof.*

**Proof 2 (Theorem 1)** *Since the $\mathcal{W}_1$ metric is defined between distribution functions, we use $\boldsymbol{v} = \mathbb{P}[\boldsymbol{v}]$ throughout the proof to denote the probability distribution over elements of the dynamics vector $\boldsymbol{v}$. In practice, this amounts to re-normalizing the representation.*

*For two dynamics to be assigned to the same cluster $j$, the following has to hold:*

$$\sum_{i=1}^{|\mathcal{V}|} \boldsymbol{v}_i \mathbf{E}_{ji}^\top > \sum_{i=1}^{|\mathcal{V}|} \boldsymbol{v}_i \mathbf{E}_{j'i}^\top,$$

$$\sum_{i=1}^{|\mathcal{V}|} \boldsymbol{v}_i' \mathbf{E}_{ji}^\top > \sum_{i=1}^{|\mathcal{V}|} \boldsymbol{v}_i' \mathbf{E}_{j'i}^\top \tag{14}$$

*for any $1 \leq j' \leq E$ s.t. $j' \neq j$. Then, adding both inequalities yields, for all $1 \leq j \leq E$*

$$
\begin{aligned}
\sum_{i=1}^{|\mathcal{V}|} \boldsymbol{v}_i \mathbf{E}_{ji}^\top + \sum_{i=1}^{|\mathcal{V}|} \boldsymbol{v}_i' \mathbf{E}_{ji}^\top &\geq \sum_{i=1}^{|\mathcal{V}|} \boldsymbol{v}_i \mathbf{E}_{j'i}^\top + \sum_{i=1}^{|\mathcal{V}|} \boldsymbol{v}_i' \mathbf{E}_{j'i}^\top \\
\sum_{i=1}^{|\mathcal{V}|} \boldsymbol{v}_i \mathbf{E}_{ji}^\top + \sum_{i=1}^{|\mathcal{V}|} \boldsymbol{v}_i \mathbf{E}_{j'i}^\top &\geq \sum_{i=1}^{|\mathcal{V}|} \boldsymbol{v}_i' \mathbf{E}_{ji}^\top + \sum_{i=1}^{|\mathcal{V}|} \boldsymbol{v}_i' \mathbf{E}_{j'i}^\top \\
\sum_{i=1}^{|\mathcal{V}|} \boldsymbol{v}_i (\mathbf{E}_{ji}^\top - \mathbf{E}_{j'i}^\top) &\geq \sum_{i=1}^{|\mathcal{V}|} \boldsymbol{v}_i' (\mathbf{E}_{ji}^\top - \mathbf{E}_{j'i}^\top) \\
\sum_{i=1}^{|\mathcal{V}|} \boldsymbol{v}_i (\mathbf{E}_{ji}^\top - \mathbf{E}_{j'i}^\top) &\geq \sum_{i=1}^{|\mathcal{V}|} \boldsymbol{v}_i' (\mathbf{E}_{ji}^\top - \mathbf{E}_{j'i}^\top)
\end{aligned}
\tag{15}
$$

*and the constraint of two vectors belonging to the same cluster $j$ becomes*

$$
\begin{aligned}
\sum_{i=1}^{|\mathcal{V}|} (\boldsymbol{v}_i - \boldsymbol{v}_i')(\mathbf{E}_{ji}^\top - \mathbf{E}_{j'i}^\top) &\geq 0 \\
\min_{1 \leq j' \leq E} \sum_{i=1}^{|\mathcal{V}|} (\boldsymbol{v}_i - \boldsymbol{v}_i')(\mathbf{E}_{ji}^\top - \mathbf{E}_{j'i}^\top) &\geq 0 \\
\min_{1 \leq j' \leq E} (\boldsymbol{v} - \boldsymbol{v}')(\mathbf{E}_j - \mathbf{E}_{j'})^\top &\geq 0
\end{aligned}
\tag{16}
$$

*Now, denote $\mathbf{E}(j) := \mathbf{E}_j$. Our constraint satisfaction problem can be written as*

$$
\min_{1 \leq j' \leq E} (\boldsymbol{v} - \boldsymbol{v}')(\mathbf{E}(j) - \mathbf{E}(j'))^\top \geq 0
\tag{17}
$$

*By comparing Eq. 7 with Eq. 17, we observe that in our case, $\mu$ is restricted to the set of vectors in $\mathbb{R}^{|\mathcal{V}|}$. Therefore, we pick $\mu \in \Gamma(\mathbf{E})$, where $\Gamma(\mathbf{E}) = \{\boldsymbol{\omega} \in \mathcal{V} : \boldsymbol{\omega} = \mathbf{E}(j,i) - \mathbf{E}(j',i), 0 \leq \boldsymbol{\omega}_i \leq 1, \cos(\boldsymbol{v} - \boldsymbol{v}', \boldsymbol{\omega}) \in [0, \pi] | 1 \leq i \leq |\mathcal{V}|, 1 \leq j' \leq E\}$. The set $\Gamma(\mathbf{E})$ is non-empty if $\max_{l,l'} ||\mathbf{E}_l - \mathbf{E}_{l'}||_\infty \leq 1$, which holds due to $\ell_p$ norm ordering and since $\mathbf{E}$ is normalized in the $\tilde{\mathbf{Q}}$ scores expression. Adopting this notation simplifies the previous expression to*

$$
\min_{\mu \in \Gamma(\mathbf{E})} (\boldsymbol{v} - \boldsymbol{v}')\mu^\top
\tag{18}
$$

*Once again, recall that $\mathbf{E}$ is normalized. Therefore, we have*

$$
\left( \frac{\boldsymbol{e}_{ij}}{||\boldsymbol{e}_j||_2} - \frac{\boldsymbol{e}_{i'j}}{||\boldsymbol{e}_j||_2} \right) + \left( \frac{\boldsymbol{e}_{ij}}{||\boldsymbol{e}_j||_2} - \frac{\boldsymbol{e}_{i'j'}}{||\boldsymbol{e}_j||_2} \right) \leq d(i, i')
\tag{19}
$$

*which equivalently can be re-stated as (for $e_i \geq e_j$ WLOG):*

$$
\begin{aligned}
\boldsymbol{e}_{ij} - \boldsymbol{e}_{i'j} &\leq \frac{||\boldsymbol{e}_j||_2}{2} d(i, i') \\
\boldsymbol{e}_{ij} - \boldsymbol{e}_{i'j} &\leq \frac{||\boldsymbol{e}_j||_2^2}{2} (\mathbf{E}_{ij} - \mathbf{E}_{ij'}) \\
\boldsymbol{e}_{ij} - \boldsymbol{e}_{i'j} &\leq \frac{||\boldsymbol{e}_j||_2}{2} (\boldsymbol{e}_{ij} - \boldsymbol{e}_{i'j}),
\end{aligned}
\tag{20}
$$

*where we take, as an example, $d(i, i') = ||\boldsymbol{e}_j||_2 (\mathbf{E}_{ij} - \mathbf{E}_{ij'})$.*

*The final expression for the sufficient condition for two dynamics embeddings to belong to the same partition is*

$$\min_{\mu \in \Gamma(\mathbf{E})} (\boldsymbol{v} - \boldsymbol{v}')\mu^\top$$
$$s.t. \ \mu(i) - \mu(i') \leq d(i, i') \tag{21}$$

*for $d(i, i') = ||\boldsymbol{e}_j||_2(\mathbf{E}_{ij} - \mathbf{E}_{ij'})$, which is similar to the Wasserstein-1 distance under $d$, i.e. $\mathcal{W}_1^d(\boldsymbol{v}, \boldsymbol{v}')$.*

*We constructed an operator similar to $\mathcal{F}(d)$ in Ferns et al. (2004). $d$ can be computed by recursively applying $\mathcal{F}(d)$ at each $\boldsymbol{v}, \boldsymbol{v}' \in \mathcal{V}$ pointwise, which is similar to what is done in CTRL. This concludes our proof and shows how our clustering procedure can be viewed as finding reward-free bisimulations.*

However, note that the exact interpretation of *reward-free bisimulation relation* depends on how $\boldsymbol{v}$ is defined. Taking $\boldsymbol{v}$ to be two consecutive timesteps of state-action pairs yields the closest possible to the original definition of bisimulation, while sampling temporal keypoints far across the trajectory will induce a different set of properties.

