# OpenReview forum: "Cross-Trajectory Representation Learning for Zero-Shot Generalization in RL"
_ICLR.cc/2022/Conference — ICLR 2022 Poster_

### Official Review · Reviewer_7TDG · 2021-11-01

**Correctness:** 3
**Technical Novelty And Significance:** 3
**Empirical Novelty And Significance:** 3
**Recommendation:** 6
**Confidence:** 3

**Main Review:**

#### **Strengths**
- The paper tackles a very important problem in the deep RL community.
- While various components that constitute CTRL were previously introduced in other works, I believe that the paper presents a novel combination of these methods and applies them in a novel problem setting.
- The proposed method is simple and can be easily plugged in to solve ZSG problems other than Procgen.
- The experiments are extensive, as the authors have compared CTRL to various baselines, including bisimulation metrics and SSL-based methods. The proposed method also outperforms quite significantly these baselines in most tasks.
- The paper is well written and easy to understand.

#### **Weaknesses**
- The major issue with the paper is the lack of discussions and explanations on why the proposed objective functions allow the encoder to recognize the behavioral similarity. Consider a scenario where a robot arm needs to pick up a particular object on a table. Each time the robot performs its task, the surface of the table will randomly change its color, which means the distribution of tasks is now the distribution of the colors. In this imaginary scenario, maximizing $\mathcal{L}_{\text{clust}} + \mathcal{L}_{\text{pred}}$ might learn an encoder that only encodes the color information, which is not what we want to achieve. If we utilize the reward information, however, we can avoid this behavior by forcing the encoder to not encode what is not useful to predict the reward. So the reward information in this case is actually helpful for learning a good representation.
- The visualization in Figure 8 in the Appendix shows that in practice CTRL does recognize the behavioral similarity. But again, I want to understand more on why this is the case, and why the encoder does not 'cheat' in maximizing the proposed objective function.

#### **Minor comments**
- In the first sentence of the second paragraph on Page 4, the distribution should be $P(\mathcal{O}, \mathcal{Z}, \mathcal{S})$. Same problem for the following sentence.
- 'repeatedly' was repeated twice in the caption for Figure 1.


**Summary Of The Paper:**

The paper addresses the problem of zero short generalization (ZSG) in RL by improving the generalization of the encoder. A new self-supervised learning objective is proposed to encourage the encoder to map behaviorally similar observations to similar representations without the use of reward signals. The overall model is named Cross Trajectory Representation Learning (CTRL), which achieves better generalization performance on the challenging Procgen benchmark suite.

**Summary Of The Review:**

In general, the paper is interesting. The proposed method is novel and while being simple, it works effectively and outperforms the baselines. However, I want the authors to discuss in more detail about why their method works well in practice while in theory it can fail.

---

> ### Author Response · Authors · 2021-11-23
> **Response**
>
> Thank you for an interesting question. There are three interconnected factors that explain why CTRL groups behaviorally similar trajectories:
>
> 1. *CTRL groups together encodings of trajectory segments, which are formed from encodings of belief states*.. Trajectories characterize the behavior of a belief state under a policy, and the assumption here is that if two belief states belong to similar trajectories **under the same or similar policy**, then these belief states are behaviorally similar.
>
> 2. *In each update epoch, CTRL operates over trajectories from very similar, or even the same, policies.* This is because in each epoch, CTRL uses data gathered by the policy currently considered by the RL algorithm (e.g., PPO). This is why the assumption in part (1) is valid throughout CTRL’s operation. Moreover, as RL training progresses, the current policy becomes better and better because the RL algo optimizes a reward-based RL loss. I.e., CTRL doesn’t ignore the reward signal -- it just uses it implicitly, by relying on the RL loss to “choose” good policies w.r.t. which CTRL computes behavioral similarity.
>
> 3. *To optimize CTRL’s clustering loss, making encodings of trajectories similar for trajectories that are behaviorally similar.* Going to your example, consider two trajectories that are identical in their action sequence and world state sequence but differ in just the table color (i.e, have a purely visual difference in the image observation sequence). This difference is very easy to ignore during encoding and map pairs of beliefs from each timestep to the same representation.
>
> Now, consider two trajectories from your example that have the same tabletop color but differ *behaviorally* -- i.e., in their sequences of actions and having substantial differences in image observations (such as different viewing angle or objects in different positions). These differences are much harder for an encoder to learn to ignore. Not impossible, but they require learning a far more complicated transformation of the trajectories.
>
> Thus, in your example, behaviorally similar but visually different trajectories (and belief states) are much more likely to be grouped together than behaviorally different but visually similar ones.

---

### Official Review · Reviewer_z5qk · 2021-11-02

**Correctness:** 4
**Technical Novelty And Significance:** 4
**Empirical Novelty And Significance:** 3
**Recommendation:** 6
**Confidence:** 3

**Main Review:**

Strengths:

1. This paper is well-written and easy to follow.
2. The proposed method of training the encoder using clustering and prediction losses seems novel to me.
3. Experimental results in all 16 tasks in the Procgen benchmark show that the proposed CTRL generally achieve zero-shot generalization.
4. The additional analysis on clustering convergence and local perceptual changes are helpful for practical usage of CTRL.

Weaknesses:
1. A lot of details are omitted from the main paper, e.g. the detailed definitions of the clustering loss and the prediction loss.
2. The proposed method does not significantly outperform baselines in many tasks. Other than bigfish, bossfight and starpilot, the improvement made by CTRL compared to baselines is relatively marginal.

Other questions:
1. Is it possible that the representation loss leads to a degenerate solution? For example, the encoder maps all observations to the same representation. I do not find a constraint to avoid degeneration.
2. What is the relation between this proposed clustering+prediction loss and other contrastive learning based methods such as CURL?
3. How does it compare to the Darla method[1]?

[1] Higgins, Irina, et al. Darla: Improving zero-shot transfer in reinforcement learning.


**Summary Of The Paper:**

This paper proposes a new zero-shot generalization method for RL that uses a novel clustering+prediction loss to learn a cross-trajectoriy encoder. The learned encoder is supposed to map behaviorally similar observations to similar representations. The algorithm generally outperforms representation RL baselines in experiment.

**Summary Of The Review:**

This paper proposes a novel idea of learning representation with a clustering+prediction loss, which makes the representation generalize to novel observations from the same task distribution. The experimental results are convincing, although a little marginal in many environments. The proposed method has some potential to help representation learning in RL and zero-shot generalization.

---

> ### Author Response · Authors · 2021-11-23
> **Response**
>
> Thank you for your comments. We provide a detailed response below:
> 1. *Regarding DARLA*: While the setting the DARLA method considers is similar to ours in that it includes a notion of training on a source MDP in the hopes of generalizing on a target MDP while we consider generalization from a train to test task distributions with some assumed shared structure. Both our methods also seek to learn a low-dimensional summary of relevant information through representation learning. However, DARLA requires a distinct initial phase where they collect data from an environment from a partially trained RL policy (for example in the appendix in section A.2.2. They collect 4 million pixel frames), and then require a separate phase where they train a beta-VAE algorithm over the collected  pixel observations to learn a latent representation which is then used for the source task. In contrast, our method is fully online and is able to do representation learning alongside reinforcement learning, and is therefore tackling a different problem setting than DARLA.
>
> 2. *Regarding CURL and other contrastive methods*: CURL is indeed a representative algorithm of the contrastive representation learning methods in deep RL. The major issue with it is that it does not take into account the temporal aspect of the problem, as it simply applies a contrastive objective between two augmented copies of the same observation. PSEs (Agarwal et al. 2021) is a contrastive method which measures policy similarity using the actions taken by two policies in two (PO)MDPs. The major drawback of PSEs is that high-entropy policies tend to generate drastically different action sequences, but which can result in identical downstream performance. For that reason, computing action-based similarities in Procgen might be misleading and lead to similarity overestimation, along with high variance.
>
> 3. *Regarding degenerate solutions*: Your concern about representation collapse is actually one of the reasons why we introduced  the predictive loss.. For N points, Sinkhorn-Knopp will produce a set of K partitions with equal amounts of data in each, due an entropy boosting term. By itself, this representation is not likely to collapse, since it has to encode information about N/K points per partition, a constraint which will be violated if all points are mapped into the same cluster. However, the clustering procedure is discriminative in nature, i.e. the mutual information between the K partitions is low. To find a good control policy via gradient descent, the representation must allow a smooth transition in between the prototypical behaviors. Hence, each prototypical behavior must contain information about the nearby (most likely) clusters, hence the convenience of using MYOW.
>
> 4. *Regarding algorithmic details*: Apologies for this confusion, there was an issue with lack of space, due to which the actual algorithm was moved to the Appendix 8.2. All algorithmic details can be found there. The clustering loss is Eq.5, while the prediction loss is Eq.6-7. We apologize for this confusion, once again.
>
> 5. *Regarding performance*: As you said, CTRL’s improvement over the presented baselines is concentrated in multiple games (with smaller improvements throughout all games). We want to highlight that DAAC had a similar improvement over PPG, concentrated mostly in bigfish, dodgeball and plunder.

---

> > ### Comment · Reviewer_z5qk · 2021-11-29
> > **Thank you for the clarification**
> >
> > Thank you for the clarification and the revision. My ratings about this paper are not considerably changed.

---

### Official Review · Reviewer_Kekc · 2021-11-02

**Correctness:** 3
**Technical Novelty And Significance:** 3
**Empirical Novelty And Significance:** 2
**Recommendation:** 3
**Confidence:** 4

**Main Review:**

**Originality and significance**:
This paper addresses zero-shot generalization, which is an important problem in RL. The idea of using unsupervised representation learning techniques to avoid overfitting in RL is not new. But the proposed method, CTRL, is somehow novel in that it uses both online clustering and cross-cluster prediction for representation learning, which has not been explored in the literature.

**Quality**:
The overall quality of the paper is below the threshold for acceptance. Below are detailed comments.
1. The setup for the ProcGen experiment presented in Section 6.1 is unclear. ProcGen has two difficulty modes and the main text is not clear about which one was used in the experiment. I guess the easy mode was used based on the number of training levels. Moreover, the 8 million steps training budget is unusual. The original ProcGen paper recommended 25 million steps for the easy mode. The choice of reporting test performance after 8 million steps of training seems arbitrary and needs further justification.
2. I have some questions regarding the results in Table 1. Although the main text says "...DAAC also exhibits good generalization performance...", DAAC in fact performs worse than PPO (6 wins, 9 loses, and 1 tie in 16 games). These results conflict with the results reported in the original paper by Raileanu and Fergus. This is a bit surprising as Raileanu and Fergus opensourced their code. One possible explanation is that the discrepancy is due to the lower training budget, but it then leads back to the first question of why using 8 million steps rather than the recommended 25 million steps. Overall, the discrepancy between the reported results and the published results makes Table 1 less convincing.
3. The experiment in Section 6.1 is very helpful. It verifies the importance of adapting the clusters online as the policy improves.
4. I have some difficulty understanding the purpose of Section 6.3. The main text says "To demonstrate the importance of identifying behavioral similarities..." But after reading this section through, all I got was a way to select the hyperparameter for the number of clusters. I will be grateful if the authors can provide some clarification.

**Clarity**:
The overall flow of the paper is easy to follow. But there are critical details missing in the method description which makes it hard to understand. Although the Sinkhorn-Knopp procedure and MYOW are existing methods, the authors should provide minimum details in the *main text* so that the readers can understand what is going on. For example, the concrete equations for computing $L_{\text{clust}}$ and $L_{\text{pred}}$ should be presented. Figure 1 can be explained in more detail. Currently, it contains unexplained notations like $\textbf{e}_{c}$.

**Minor issues & typos**:
* The references seem truncated. For example, all the citations to Zhang _et al_ are not shown in the references.
* The citation to Kendall _et al_ in the 3rd line of Section 1 should use \citep.
* The 7th line on page 3: "bisimation" --> "bisimulation".
* The 4th last line on page 5: the citation to Raileanu and Fergus should use \citep.
* Bullet point 2 in Definition 1: if $c$ is a state, what does $s' \in c$ mean? Is it a typo?
* The line below Definition 1: the citation to Ferns _et al_ should use \citet.

**Summary Of The Paper:**

This paper addresses the zero-shot generalization (ZSG) problem in reinforcement learning (RL). Specifically, it focuses on learning state representations that can generalize well. To this end, the authors propose a self-supervised learning method, cross trajectory representation learning (CTRL), that captures behaviorally similarity in the trajectory representations. The CTRL objective has two components, one for online clustering and the other one for cross-cluster prediction. CTRL is shown to be connected to bisimulation metrics in RL. The empirical study in the ProcGen benchmark shows that CTRL performs better than existing methods in the literature. Further experiments demonstrate the importance of adaptive online clustering and shed light on how to choose the number of clusters as a hyperparameter.

**Summary Of The Review:**

I think this paper is mainly held back by two factors. First, critical details are missing in the main text which makes the proposed method hard to understand. Second, I have some questions regarding the main empirical results that may undermine the empirical significance of this work. Therefore, I don't think this paper is ready to publish yet.

---

> ### Author Response · Authors · 2021-11-23
> **Response**
>
> Thank you for your comments. We are providing answers to your questions below:
> 1. *Regarding experimental setup*: We agree with you on the matter of the experimental setting being different from the original 25M steps, and we apologize for any confusion which our formulation might have brought. First of all, our exact experimental setting is: train on 200 levels of ‘easy’ in Procgen, train on 8M frames, and then measure test performance on the entire distribution of ‘easy’. The 8M frames setting is taken from this paper: “Measuring Sample Efficiency and Generalization in Reinforcement Learning Benchmarks: NeurIPS 2020 Procgen Benchmark”, which allows testing algorithms on a reduced computational budget (akin to the 100k data-efficient Atari benchmark vs the full 200M). We have clarified this in the updated paper version.
>
> 2. *Regarding baselines*: It is widely known that there are issues with reproducing Procgen + PPO performance outside of their native codebase in Tensorflow. While the DAAC code is in PyTorch, its running time is much slower than that of the original Procgen code. We have been in touch with the authors of DAAC to appropriately port their code onto our codebase. The relative ordering of PPO vs DAAC performance comes from the fact that our results are on 8M, which produces a different ordering of baselines from 25M, but can be run in a much more computationally efficient way (especially since DAAC has a total of 10 epoch updates and two IMPALA forward passes, as opposed to PPO’s 3 epoch updates and one IMPALA forward pass).
>
> 3. *Regarding the composite Ising model*: We apologize if this section caused any confusion. Its goal was as follows: given a toy sequential decision-making scenario (in fact, it is a multi-armed contextual bandit problem) where the structure clearly factorizes into k distinct behaviors parametrizes by some unknown parameter, can learning the underlying structure help obtain higher returns on the downstream task of selecting the arm with the highest reward? The answer, in this simple task, is yes, as a higher silhouette score (clustering performance) corresponds to a higher performance score (downstream task performance). The practical difficulty is in transposing this approach onto complex domains such as Procgen, since we do not know the underlying connectivity structure of the task. The best we can hope to do is monitor the silhouette score, which has its own drawbacks, as we discuss in our response to R1.

---

### Official Review · Reviewer_DS9R · 2021-11-03

**Correctness:** 3
**Technical Novelty And Significance:** 3
**Empirical Novelty And Significance:** 4
**Recommendation:** 6
**Confidence:** 4

**Main Review:**

I found the paper interesting, but rather hard to understand what it really contributed and how several decisions came about. It is really only after I read MYOW and SwAV that I could understand several model architecture and loss choices, as the main text doesn’t cover them well enough and the Appendix is quite lacking. It also appeared to me that the proposed model is really close to MYOW and that fact wasn’t made as clear as I would have liked.


Additional comments and questions:

1. The introduction and background are very clear and set up the problem well. The literature review is thorough and recent, and the overview of the algorithm in section 4 is also clear.
2. My main issue was in understanding several model choices, when looking at Figure 1 especially. After looking at MYOW and SwAV, one can see how the overall model is basically MYOW with a different sequence of networks (a simpler one, which appears more plausible than MYOW to me, although see the next point) combined cleverly with SwAV’s soft-clustering model. CTRL also removes all multi-views aspects of these works, instead operating on trajectories directly.
   1. It would have been much clearer to spell out these differences and what you did explicitly, given I would still consider that CTRL is significantly different from either of these two previous works.
   2. One particular point which was confusing was the decision to make latent space be similar to adjacent clusters. This is taken directly from MYOW, but wasn’t clearly expressed here. It also appears slightly odd as there is no enforcement of a “global coverage” across the clusters (i.e. there is no extra repulsion term to far-away clusters), which to me would indicate that this method will have a chance to collapse (although one can be lucky by using target networks, ala BYOL).
   3. Similarly, the choice to have a cascade of networks creating the two embedding for the clustering is rather surprising (and I would expect it to collapse to degenerate solutions given you’re not applying them in swapped ways like in SwAV).
   4. The main text talks about trajectories, but T=2 if the Appendix is to be trusted. Hence in practice you are only clustering transitions. It would have been easier to present it that way (and the connection to bisimulation metrics would have been clearer).
3. Many details aren’t in the main text, and the Appendix is equally light in details:
   1. What is your encoder? Does it receive a history / have a recurrent state? Appendix 8.2 presents it as if it is instantaneous, which is quite a limitation.
   2. How is FiLM used exactly? There are infinitely many ways to add it in.
   3. It is very hard to follow the exact sequence of v/w/v’/w’ and how they are computed. One needs to carefully read SwAV to build an intuition, but this shouldn’t be that hard. No details are provided on what these are, apart from “\psi_clust is a RNN” and “\theta_clust is a MLP”
   4. MYOW explicitly used predictors everywhere and were never regressing towards the latent used to cluster (i.e. `y = f_theta(x)` in their paper. See Figure 1, the augmented and clustering loss are applied on z and v, respectively.). Instead, you directly force `w’` to be predictable of `v’_c_i`. It would be valuable to discuss why you believe this is more appropriate (MYOW’s argue that this worked worst for them).
   5. It would be quite valuable to bring back some of Section 8.2 into the main text, and add many more details to clarify what the model really is, I don’t believe this is reproducible in this current state.
4. Table 1 is great and I would again like to flag that the number of baselines available are clearly to a great standard. This must have been hard to do well, so congratulations!
   1. One caveat is that you are only reporting Evaluation performance on held out tasks (which is good, that was your metric), but I always would have appreciated seeing the performance on the training distribution. This would make it clearer if the effect is about improved generalization, or about the policies themselves being better or worse for particular baselines/ablations.
5. Section 6.2 is quite anecdotal, and it isn’t clear that the effect discussed is that significant?
6. Section 6.3 feels out of place and is quite underwhelming after having covered the results on Procgen.
   1. It is a very peculiar toy situation, which isn’t that standard, and I’m not entirely sure what it’s trying to demonstrate in its current state.
   2. Can you use the same silhouette score change to tune the number of clusters in Procgen too? Given it’s a post-hoc analysis, this doesn’t really appear like a real solution to this problem.
7. Figure 4 in the Appendix has T=3 and T=5, but they show no difference? There is also no discussion of Figure 4, the section in 8.3 appears empty.
8. As a small point, you keep referring to “views”, but this is overloaded and isn’t as appropriate as you do not actually have augmentations like in both previous work? Once again makes it hard to follow but should be easy to fix.
9. You might want to add a citation to Van der Pol et al 2020 (https://arxiv.org/abs/2002.11963) for the bisimulation work, as they had a slightly different take than Zhang et al 2021.


**Summary Of The Paper:**

This paper proposes an extension of the MYOW (Azabou et al 2021) self-supervised learning technique, combining it with SwAV (Caron et al 2021)’s method to perform online clustering, and adapt it for RL to assess generalisation performance on Procgen.
It compares against several recent baselines (DBC, PSE, CURL, Proto-RL and DIAYN), and outperforms them consistently on the tasks assessed (albeit by a small margin, depending on the reward scale of Procgen which I’m not extremely familiar with).

**Summary Of The Review:**

Overall, I think this is nice paper, with good baselines and clear results, however, its presentation right now is quite lacking and hence I feel it may need some work before being ready for publication.

---

> ### Author Response · Authors · 2021-11-23
> **Response**
>
> Thanks for the super-detailed review! First of all, we agree that some of the presentation issues that you mention are indeed present, and we provide our clarifications below:
> 1. *Regarding collapse issues*: We apologize for the confusion, but the MYOW predictors do indeed use EMA targets, just like you said and as is the case in BYOL and others. Additionally, they are stop-graded which, in itself, has also been shown sufficient to prevent collapse (shown in SimSiam method).
>
> 2. *Regarding the choice of encoder and T*: Since our setting is in fact a POMDP, we are concerned with learning latent state representations of a belief state (i.e. entire history of pixel observations). However, the original Procgen paper has shown that using a recurrent mechanism is not necessary to obtain good performance on the benchmark. If we were in an MDP, then indeed, using T=2 would have boiled down exactly to performing a clustering step over model dynamics. However, using partial views of the environments will produce something less informative. Ultimately, CTRL learns a mapping from pixels to a latent vector s.t. Observations with similar dynamics have nearby representations and potentially similar actions, as the controller is a simple linear layer on top of the representation. On a high level, CTRL ensures the encoder is aware of multiple “behavior scenarios” in the environment and maps them to a common action.
>
> 3. *Regarding FiLM*: Given a sequence of observation-action pairs $\{o_t,a_t,o_{t+1},a_{t+1},..\}$, we first encode all observations into latent vectors, which are then conditioned on their respective actions using FiLM, i.e. $(o_t,a_t)$, $(o_{t+1},a_{t+1})$, etc. The resulting vectors are then concatenated into a latent signature trajectory vector of dimension $T \times hidden$, which is then fed to the clustering and prediction heads.
>
> 4. *Regarding silhouette score and post-hoc tuning*: this is a great remark, and one of the reasons we provided the silhouette metric. While, as we have shown in the appendix plots, there is some dependence between the silhouette score and the test-time returns on some games, there is also the danger of using the silhouette score as a proxy. Naturally, cluster assignment becomes harder with an increasing number of clusters due to curse of dimensionality, and therefore partitions with lower numbers of clusters will often lead to higher silhouette scores. However, since the cluster centroids correspond to representative instances of belief state dynamics, lower granularity in that space might hurt performance. Nevertheless, the silhouette score can still be used in post-hoc analyses to determine the optimal number of prototypical behavior by considering a dependence graph of silhouette vs test-time results.
>
> 5. *Regarding terminology and previous works*: We agree that the terminology can be sometimes confusing due to abuse of notation. We also will include the Van der Pol paper in the citation list.

---

> > ### Comment · Reviewer_DS9R · 2021-11-30
> > **Thank you for the response**
> >
> > Thank you for your hard work in getting a new revision of the paper in and addressing most of my comments.
> >
> > 1. The rewrite added much more details on algorithmic choices and details of relations to MYOW and SwAV. This makes it much clearer to follow and I believe this improves the paper quite a lot, at the very least by providing a clear presentation of how one could combine these methods.
> > 2. Thanks for adding the training curves Figure 10 and evaluation curves Figure 11. These help give more insights into Table 1. Unfortunately, it does appear that the difference between methods isn't extremely clear across most levels considered. However, given there is a clear effect visible and this would be valuable insights for other researchers, I'd be willing to support publishing in this state despite these issues (Deep RL is what it is currently after all).
> > 3. For FiLM I actually wanted to know the exact modulation you used (i.e. only biases or did you learn the scaling too? A single one or one per channel, etc). This would be clearer from the code, but I'd encourage you to provide more details of the exact architectures/math used (same for all other modules you used, pointing to other papers isn't as self-contained) in the Appendix to make all of these confusions disappear.
> >
> > I've increased my score to reflect the improvements to the paper.

---

### Decision · Program_Chairs · 2022-01-20

**Decision:**

Accept (Poster)

**Comment:**

Meta Review of Cross-Trajectory Representation Learning for Zero-Shot Generalization in RL

This work investigates a zero-shot generalization method for RL based on an online-clustering adapting it to RL. The intuition of this approach (called Cross Trajectory Representation Learning, CTRL) is that the self-supervised objective used will encourage the encoder to map behaviorally similar observations to similar representations without the use of reward signals. The authors performed experiments on the 16 procgen tasks, and compared it against several baselines (DBC, PSE, CURL, Proto-RL and DIAYN). The performance is generally better against baselines, but what I like about it is that a new approach to achieve such performance is proposed.

The scores were generally good (6, 6, 6, 3), and the 6's are overall positive with the work (both in the writing, breadth of experiments). Reviewer Kekc, who gave a score of 3, maintained their score, despite acknowledging the authors' responses. The main outstanding issue from Kekc is that they believe the paper should stick with the original 25M step protocol (with a larger training budget), rather than 8M steps. If that's the main issue for this paper to not be accepted into ICLR 2022, I feel this can be adequately addressed for the camera ready version. (Please note that while I disagree with the final score of 3 that Kekc gave, I find their review to be highly informative and useful, and would like to acknowledge Kekc for their input and discussion).

Based on the discussion and the reviews, and with the context behind the score of 3, I would like to be on the side of recommending this paper for acceptance, and urge strongly for the authors to conduct the 25M experiments as reviewer Kekc suggested (as Kekc also noted, the training curves are still going up, so just train them for a longer time). Even if the final results are not as good as the 8M, that's fine, just include them in the final camera ready version, since I believe this work to meet the bar, offers good insights into RL generalization, has a good breadth of experiments and baselines, and will be of great interest to the broader RL community.